# Endogenous p53 expression in human and mouse is not regulated by its 3′UTR

**Sibylle Mitschka, Christine Mayr\***

Cancer Biology and Genetics Program, Memorial Sloan Kettering Cancer Center, New York, United States

**Abstract** The *TP53* gene encodes the tumor suppressor p53 which is functionally inactivated in many human cancers. Numerous studies suggested that 3′UTR-mediated p53 expression regulation plays a role in tumorigenesis and could be exploited for therapeutic purposes. However, these studies did not investigate post-transcriptional regulation of the native *TP53* gene. Here, we used CRISPR/Cas9 to delete the human and mouse *TP53/Trp53* 3′UTRs while preserving endogenous mRNA processing. This revealed that the endogenous 3′UTR is not involved in regulating p53 mRNA or protein expression neither in steady state nor after genotoxic stress. Using reporter assays, we confirmed the previously observed repressive effects of the isolated 3′UTR. However, addition of the *TP53* coding region to the reporter had a dominant negative impact on expression as its repressive effect was stronger and abrogated the contribution of the 3′UTR. Our data highlight the importance of genetic models in the validation of post-transcriptional gene regulatory effects.

## Introduction

The transcription factor p53 coordinates the cellular stress response. p53 regulates expression of genes involved in cell cycle control, DNA repair, apoptosis, metabolism, and cell differentiation (*Kastenhuber and Lowe, 2017*), thus allowing cells to adequately respond to various stimuli. In contrast to most other stress sensors, p53 is not dependent on de novo transcription as the *TP53* gene is continuously transcribed even in unstressed cells. This mechanism reduces the time of delay between signal detection and downstream responses. To avoid activation of p53 in unstressed cells, p53 protein levels are controlled by rapid protein degradation induced by MDM2 (*Haupt et al., 1997*; *Kubbutat et al., 1997*).

Reduced levels or insufficient p53 activity are major risk factors for the development of cancer and more than half of all human cancers exhibit diminished p53 expression or function (*Kastenhuber and Lowe, 2017*). In contrast, hyperactive p53 has been linked to impaired wound healing, obesity, and accelerated aging (*Rufini et al., 2013*). These phenomena highlight the importance of p53 protein abundance and activity regulation in human health.

For many years, the 3′ untranslated region (3′UTR) of the *TP53* mRNA has been a widely studied element of p53 expression regulation. 3′UTRs are essential for facilitating pre-mRNA processing. In addition, they can also recruit microRNAs (miRNAs), RNA-binding proteins, and lncRNAs which can modulate mRNA stability and protein translation (*Tian and Manley, 2017*; *Mayr, 2019*). The human *TP53* 3′UTR contains experimentally characterized binding sites for 23 miRNAs, 1 lncRNA, and 6 RNA-binding proteins (*Haronikova et al., 2019*). A large number of experiments demonstrated the repressive nature of the *TP53* 3′UTR using synthetic reporter assays under steady-state conditions (*Table 1*; *Haronikova et al., 2019*). In addition, the *TP53* 3′UTR was shown to facilitate an increase in p53 translation after genotoxic stress (*Fu and Benchimol, 1997*; *Mazan-Mamczarz et al., 2003*; *Chen and Kastan, 2010*). Together, this large body of work strongly suggested that miRNAs and RNA-binding proteins prevent p53 hyperactivation under normal conditions and help to upregulate

**\*For correspondence:**
mayrc@mskcc.org

**Competing interests:** The authors declare that no competing interests exist.

**Table 1.** Previously reported evidence of miRNAs, lncRNAs, and RNA-binding proteins that target the *TP53* 3'UTR.
Interactors of the human *TP53* mRNA mapping to the last exon

| Name | Type | Binding region (NM_000546.6) | Affected in dUTR allele? | Experiments | References (PMID) | Average PhyloP score |
|---|---|---|---|---|---|---|
| miR-1228–3 p | miRNA | 1422–1428 | yes | LRA, RT-qPCR, IHC, WB | 25422913 | 1.003 |
| miR-125a-5p | miRNA | 2044–2063 | yes | LRA, NB, RT-qPCR, WB | 19818772 | −0.120 |
| miR-125b-5p | miRNA | 2043–2064 | yes | LRA, ISH, RT-qPCR, WB | 19293287, 21935352, 27592685 | −0.105 |
| miR-1285–3 p | miRNA | 2113–2134 | yes | LRA, RT-qPCR, WB | 20417621 | −0.061 |
| miR-150–5 p | miRNA | 1568–1580 | yes | LRA, WB | 23747308 | −0.323 |
| miR-151a-5p | miRNA | 2304–2325 | yes | LRA, ChIP-seq, RT-qPCR, WB | 27191259 | −0.053 |
| miR-200a-3p | miRNA | 2269–2291 | yes | LRA, WB | 23144891 | −0.039 |
| miR-24–3 p | miRNA | 2352–2374 | yes | LRA, IHC, RT-qPCR, WB | 27780140 | 0.018 |
| miR-25–3 p | miRNA | 1401–1423 | yes | LRA, RT-qPCR, WB | 20935678 | 0.438 |
| miR-30d-5p | miRNA | 1596–1618 | yes | LRA, RT-qPCR, WB | 20935678 | −0.432 |
| miR-375 | miRNA | 1462–1483 | yes | LRA, Flow, RT-qPCR, WB, IF | 23835407 | −0.595 |
| miR-663a | miRNA | 1260–1281 | no (in CDS) | LRA | 27105517 | 1.520 |
| miR-504 | miRNA | 2059–2066, 2387–2395 | yes, no | LRA, RT-qPCR, WB | 20542001 | 0.130 0.203 |
| miR-92 | miRNA | 1417–1422 | yes | LRA, WB | 21112562 | 0.102 |
| miR-141 | miRNA | 2285–2290 | yes | LRA, WB | 21112562 | −0.031 |
| miR-638 | miRNA | 1381–1404 | yes | LRA, WB, IP | 25088422 | 0.313 |
| miR-3151 | miRNA | 1337–1354 | yes | LRA, WB, RT-qPCR | 24736457 | −0.329 |
| miR-33 | miRNA | 1957–1980 | yes | LRA, WB | 20703086 | 0.138 |
| miR-380–5 p | miRNA | 1909–1936, 1943–1974 | yes, yes | LRA, WB | 20871609 | 0.121 0.089 |
| miR-19b | miRNA | 1712–1734 | yes | LRA, WB | 24742936 | 0.402 |
| miR-15a | miRNA | 2394–2414 | no | LRA, WB | 21205967 | 0.045 |
| miR-16 | miRNA | 2394–2415 | no | LRA, WB | 21205967 | 0.015 |
| miR-584 | miRNA | 1263–1284 | no (in CDS) | LRA, WB, IP | 25088422 | 1.646 |
| WIG1 | RBP | 2064–2106 | yes | LRA, IP, RT-qPCR | 19805223 | −0.071 |
| PARN | RBP | 2071–2102 | yes | LRA, EMSA, IP, RT-qPCR | 23401530 | −0.097 |
| CPEB1 | RBP | 2458–2500 | no | IP, RT-PCR | 19141477 | 1.654 |
| RBM38 (RNPC1) | RBP | 2064–2106 | yes | EMSA, IP, RT-PCR, Polysome gradient | 21764855, 24142875, 25823026 | −0.071 |
| RBM24 | RBP | 2064–2106 | yes | LRA, EMSA, IP, RT-qPCR, | 29358667 | −0.071 |
| HUR | RBP | 2064–2106, 2393–2412, 2458–2505 | yes, yes, no | LRA, EMSA, WB, RT-qPCR | 12821781, 14517280, 16690610, 18680106 | −0.071 0.007 1.496 |
| 7SL | lncRNA | 2107–2149, 2194–2240, 2269–2301, 2307–2362 | yes, yes, yes, yes | LRA, IP, WB | 25123665 | −0.158 −0.110 0.015 −0.015 |
| miR-92a-3p | miRNA | 1646–1666 | yes | LRA, WB | 22451425 | −0.122 |
| TIA1 | RBP | 1426–1442 1702–1731 | yes, no | LRA, iCLIP | 28904350 | 0.066 0.715 |
| Hzf | RBP | 1345–1395 1529–1574 | yes, yes | LRA, EMSA, WB, IP, RT-qPCR, Polysome gradient | 21402775 | −0.450 0.156 |

LRA: luciferase reporter assay; WB: western blot; IP: co-immunoprecipitation assay; RT-qPCR: quantitative reverse transcription PCR; NB: northern blot; IHC: immunohistochemistry; ISH: In situ hybridization; EMSA: electromobility shift assay.

p53 protein translation after exposure to genotoxic stress (*Fu and Benchimol, 1997*; *Mazan-Mamczarz et al., 2003*; *Chen and Kastan, 2010*).

These findings also suggested that deregulation of p53 expression through mechanisms involving its 3′UTR could be a major disease driver as well as a potential target for treatment (*Kasinski and Slack, 2011*; *Hermeking, 2012*). In particular, miRNAs targeting the *TP53* mRNA were previously established as putative gatekeepers that prevent p53 hyperactivation and some of these miRNAs are also elevated in cancer, e.g. miR-504, miR-30d, and miR-125 (*Hu et al., 2010*; *Li et al., 2012*; *Banzhaf-Strathmann and Edbauer, 2014*). This has sparked an interest in exploiting 3′UTR-mediated expression regulation for therapeutic applications using novel miRNA-based approaches (*Kasinski and Slack, 2011*; *Hermeking, 2012*). However, the role of the 3′UTR in regulating p53 expression regulation has not been investigated using the endogenous *TP53* mRNA and the potential benefit of these approaches remains unknown.

In order to evaluate the contribution of the *TP53* 3′UTR to p53 expression regulation, we removed the *TP53* and *Trp53* 3′UTRs at orthologous human and mouse gene loci while keeping mRNA processing intact. In HCT116 cells and in mouse tissues, we did not observe 3′UTR-dependent differences in p53 mRNA or protein levels under normal conditions or after DNA damage. These results suggest that the 3′UTR of *TP53* typically has no function in regulating p53 abundance in its endogenous context. Consistent with previous reports, we did observe that the *TP53* 3′UTR in isolation represses reporter expression in reporter assays. However, adding the *TP53* coding region to the reporters entirely eliminated 3′UTR-dependent expression differences. Together, our results suggest that the *TP53* coding region masks the repressive effects of the 3′UTR in the endogenous *TP53* mRNA.

## Results

### Removal of the endogenous 3′UTR does not alter *TP53* mRNA expression

3′UTRs are known to perform two general functions: They contain regulatory elements that enable mRNA 3′ end processing and they harbor elements that allow post-transcriptional gene regulation (*Matoulkova et al., 2012*). 3′ end processing is an essential part of mRNA maturation and is facilitated by the poly(A) signal together with surrounding sequence elements that bind the polyadenylation machinery (*Martin et al., 2012*). Based on genome-wide mapping of polyadenylation factor binding sites by CLIP, we consider the 3′UTR sequence that is located up to 100–150 nucleotides upstream of the cleavage site as essential for pre-mRNA processing (*Martin et al., 2012*). As the human *TP53* 3′UTR has a total length of about 1200 nucleotides, the additional sequence may enable regulatory functions by recruiting miRNAs and RNA-binding proteins. Indeed, the vast majority of previously characterized binding sites for miRNAs and RNA-binding proteins are located in the upstream, non-essential part of the *TP53* 3′UTR (*Figure 1A*, *Table 1*).

To investigate the role of the endogenous human *TP53* 3′UTR in post-transcriptional p53 regulation, we used a pair of CRISPR/Cas9 guide RNAs to delete the non-essential part of the 3′UTR in HEK293 cells and in the human colon carcinoma cell line HCT116, an established model for investigating p53-dependent functions (*Figure 1A*, blue, *Figure 1—figure supplement 1A*). The homozygous 3′UTR deletion, called ΔUTR (dUTR), removed 1048 nucleotides, corresponding to 88% of the 3′UTR in wild-type (WT) cells. The deletion affected almost all previously reported binding sites for regulatory miRNAs, lncRNAs, and RNA-binding proteins (*Figure 1A*, *Table 1*). We confirmed intact 3′ end processing of the mRNA by Northern blot analysis and observed expression of the expected shorter *TP53* mRNA in dUTR cells (*Figure 1B*). As controls for all experiments, we used the HCT116 parental cell line and control (Ctrl) clones that underwent the genome editing procedure but have intact 3′UTRs (*Figure 1—figure supplement 1A*). We confirmed the presence and absence of the *TP53* 3′UTR in all HCT116 WT clones and dUTR clones by RT-qPCR using a primer pair in the deleted part of the 3′UTR (*Figure 1C*). We observed that one control clone showed slightly increased *TP53* mRNA level compared to the parental WT cell line (*Figure 1C*). We then used a primer pair in the coding region and measured *TP53* mRNA level in all control cells and the dUTR

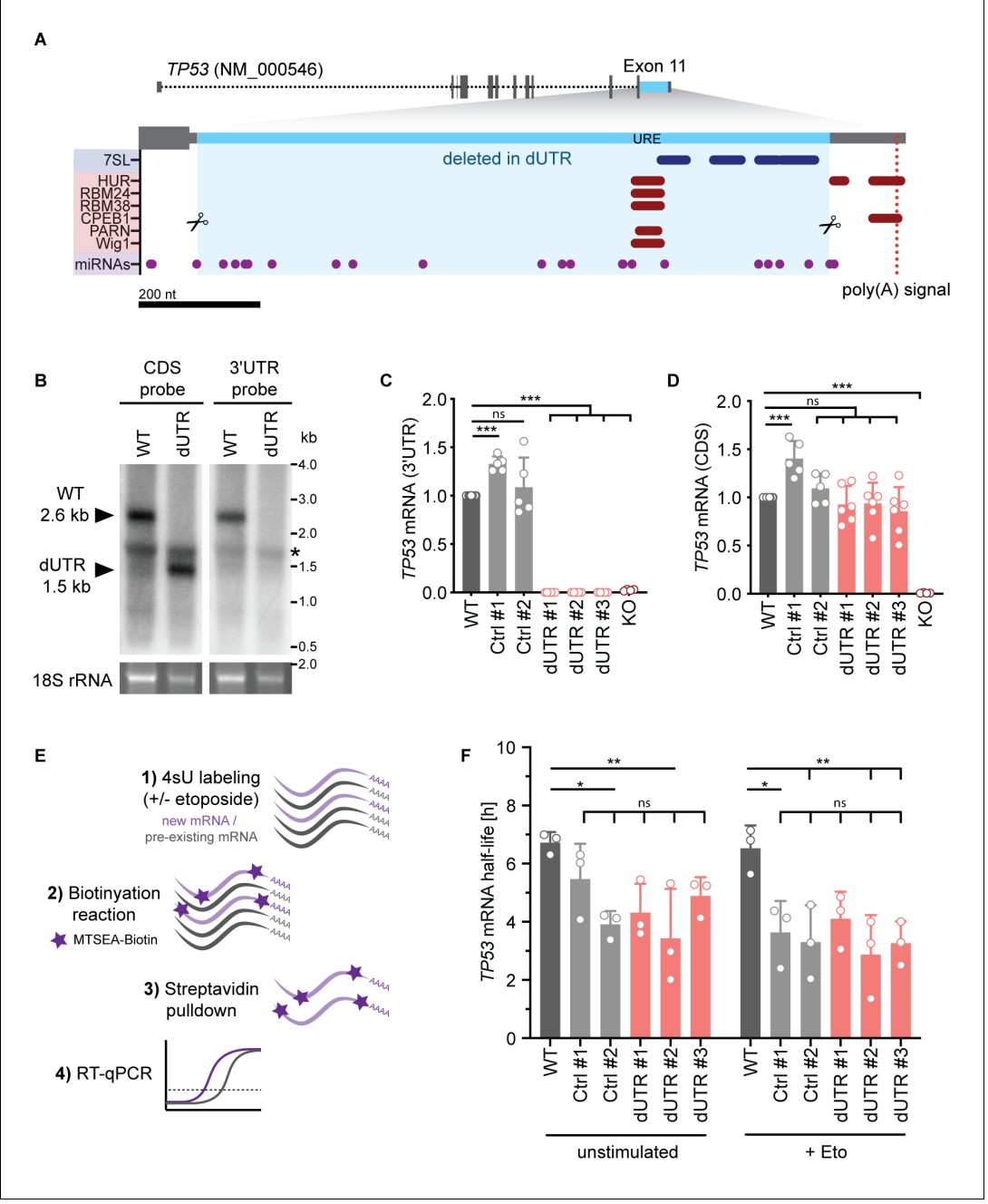

**Figure 1.** Deletion of the endogenous 3'UTR does not alter *TP53* mRNA level in human cells. (**A**) Schematic of the human *TP53* gene. The sequence deleted in dUTR cells is shown in blue. Tracks of binding sites for miRNAs, RNA-binding proteins, and lncRNAs are depicted below (see also ***Table 1***). URE, U-rich element. CRISPR/Cas9-mediated deletions at the nucleotide level are shown in ***Figure 1—figure supplement 1***. (**B**) Northern blot analysis of *TP53* mRNA from WT and dUTR HEK293 cells. A probe that hybridizes to the *TP53* coding region (CDS) reveals expression of a shortened *TP53* mRNA in dUTR cells. The size difference is consistent with the length of the CRISPR/Cas9-induced deletion. A probe designed to bind the *TP53* 3'UTR does not produce a signal in the mRNA of dUTR cells, confirming deletion of this sequence element. The band of 18S rRNA is used as a loading control. * indicates an unspecific band originating from ribosomal RNA. (**C**) *TP53* mRNA expression measured by RT-qPCR with a primer pair located in the 3'UTR in the indicated samples derived from HCT116 cells. KO, HCT116 *TP53*-/- cells. Data are shown as mean +s.d. of n = 5 independent experiments after normalization to *GAPDH*. Statistical analysis using ANOVA and Tukey's post-hoc test with ***p<0.001, ns, not significant. (**D**) *TP53* mRNA expression measured by RT-qPCR with a primer pair located in the CDS in the indicated samples derived from HCT116 cells. Data are shown as mean +s.d. of n = 5 independent experiments after normalization to *GAPDH*.

*Figure 1 continued on next page*

*Figure 1 continued*

Statistical analysis using ANOVA and Tukey's post-hoc test with ***p<0.001, ns, not significant. (**E**) Experimental setup to estimate *TP53* mRNA half-life. Metabolic labeling with 4-thiouridine (4sU) was used to enrich newly transcribed mRNAs. The newly transcribed RNAs were thiol-alkylated and biotinylated, followed by Streptavidin pull-down. Their relative abundance was measured using RT-qPCR. (**F**) *TP53* mRNA half-life obtained by 4sU labeling as described in (E) is shown for the indicated samples derived from HCT116 cells in the presence or absence of etoposide for 4 hr (Eto, 20 µM). Shown is mean + s.d. from n = 3 independent experiments. Statistical analysis using ANOVA and Tukey's post-hoc test with *p<0.05, **p<0.01, ns, not significant.

The online version of this article includes the following figure supplement(s) for figure 1:

**Figure supplement 1.** Generation and characterization of *TP53* dUTR human cell lines.

clones and observed similar *TP53* mRNA level (*Figure 1D*). This was also true for HEK293 cells carrying the homozygous dUTR deletion (*Figure 1—figure supplement 1B and C*). This indicates that removal of the endogenous 3′UTR does not influence p53 steady-state mRNA levels.

Notably, mRNA levels are determined by transcription and degradation rates and both pathways could be affected by the deletion of the *TP53* 3′UTR. We measured *TP53* mRNA half-lives in the absence and presence of DNA damage in all cell lines. To do so, we used 4-tiouridine (4sU) labeling followed by enrichment of newly transcribed mRNAs (*Figure 1E*; *Russo et al., 2017*). Among all clones, regardless of WT or dUTR genotype, we observed no difference in *TP53* mRNA half-life (*Figure 1F*). However, all clones exhibited a slightly reduced *TP53* mRNA half-life in comparison to the parental WT cell line (*Figure 1F*). This might be caused by metabolic changes occurring during clonal expansion of the cell lines. Overall, our data suggest that the 3′UTR does not impact *TP53* mRNA level or half-life.

## The endogenous 3′UTR is not involved in regulating p53 protein levels in steady state or after genotoxic stress

Next, we measured p53 protein levels in WT and dUTR cells using western blot analysis. We did not detect a significant difference in p53 protein level between WT and dUTR cells in steady-state conditions (*Figure 2A and B*). We also investigated the expression of shorter p53 protein isoforms that can be generated by alternative splicing or through alternative translation initiation. A prior study had proposed an interplay between *cis*-regulatory elements in the *TP53* 3′UTR and usage of a downstream start codon resulting in a p53 protein isoform lacking one of the N-terminal transactivation domains (*Katoch et al., 2017*). Importantly, the region that has been implicated in this process is deleted in our *TP53* dUTR allele. Using a different monoclonal p53 antibody with broad isoform specificity, we found no differences in the expression pattern in p53 protein isoforms between dUTR and WT cells (*Figure 2—figure supplement 1A and B*). This suggests that under steady-state conditions, the *TP53* 3′UTR is not necessary for the regulation of p53 protein levels.

Next, we investigated the possibility that the *TP53* 3′UTR might have a role in stress-dependent p53 protein expression regulation. While *TP53* mRNA expression does not change upon DNA damage, upregulation of p53 protein expression is achieved through increased translation rates and lower protein turnover (*Kumari et al., 2014*). Previous studies had suggested a role of the 3′UTR in the upregulation of p53 translation after exposure to genotoxic stress (*Fu and Benchimol, 1997*; *Mazan-Mamczarz et al., 2003*; *Chen and Kastan, 2010*). To assess stress-induced p53 expression regulation in dUTR cells, we treated cells with the topoisomerase inhibitor etoposide which is known to upregulate p53 protein expression by causing DNA double-strand breaks. We found that concentration-dependent upregulation of p53 protein expression was similar in parental, control clone, and dUTR cells (*Figure 2C and D*). In addition, p53 levels analyzed over a period of 2 days revealed similar p53 expression kinetics (*Figure 2E and F*). Finally, we tested additional stress stimuli including 5-fluorouracil (a thymidylate synthase inhibitor) or UV irradiation. These treatments robustly upregulated p53 protein across all cell lines, but with no significant differences in p53 expression between WT and dUTR cells (*Figure 2G and H*). We therefore concluded that the endogenous *TP53* 3′UTR is not required for p53 expression regulation neither in steady state nor after DNA damage.

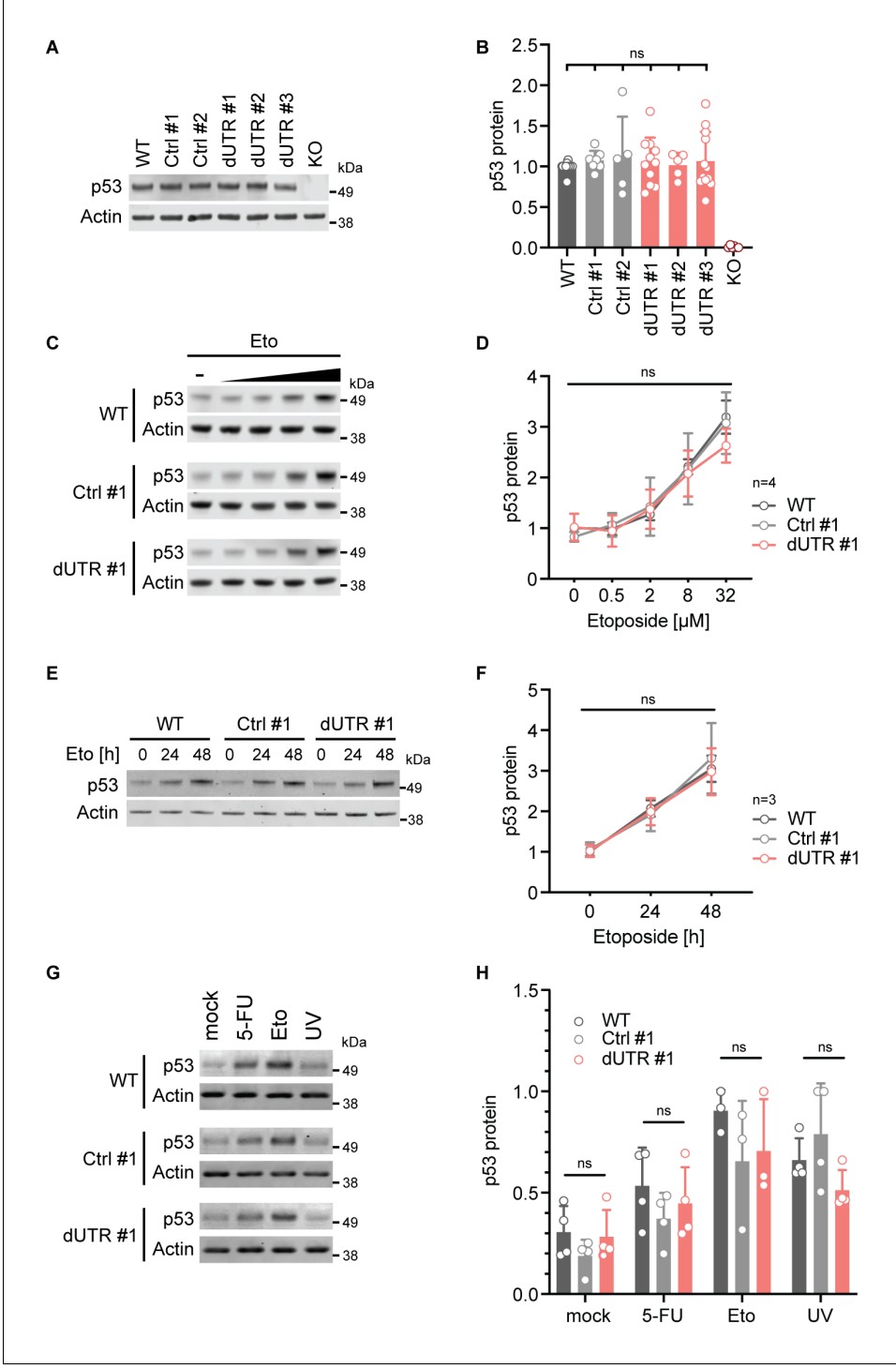

**Figure 2.** Deletion of the endogenous *TP53* 3′UTR does not alter p53 protein level in steady state or after genotoxic stress. (**A**) Representative immunoblot showing steady-state p53 protein expression in the indicated samples derived from HCT116 cells. Actin serves as loading control. See **Figure 2—figure supplement 1** for additional information on p53 isoform expression. (**B**) Quantification of immunoblot data for steady-state p53 protein level. p53 expression data were normalized to Actin and the WT parental cell line. For each sample at least five biological replicates were measured. Statistical analysis using ANOVA. ns, not significant. (**C**) Representative immunoblots showing p53 protein levels after 4 hr of etoposide (Eto) treatment (0–32 μM) in WT, Ctrl clone #1, and dUTR clone #1 derived from HCT116 cells. Actin serves as loading control. (**D**) As in (**C**). Quantification of p53 protein expression from n = 4 independent experiments is shown as mean + s.d. Statistical analysis of cell line

*Figure 2 continued on next page*

*Figure 2 continued*

effect using ANOVA. ns, not significant. See *Figure 2—source data 1* for raw data. (E) Representative immunoblot of samples from WT parental, Ctrl clone #1, and dUTR clone #1 derived from HCT116 cells that were treated with 0.5 μM Eto for 0, 24, or 48 hr (h). Actin serves as loading control. (F) As in (E). Quantification of p53 protein expression from n = 3 biological replicates is shown as mean +s.d. Statistical analysis of cell line effect using ANOVA. ns, not significant. See *Figure 2—source data 1* for raw data. (G) As in (C), but cells were treated with 20 μM etoposide (Eto), 40 μM 5-fluorouridine (5-FU), or 50 J/m$^2$ UV. Actin serves as loading control. (H) As in (G). Quantification of p53 protein expression after drug treatment. For each drug at least three biological replicates were measured. Shown is mean + s.d. Statistical analysis using ANOVA. ns, not significant.

The online version of this article includes the following source data and figure supplement(s) for figure 2:

**Source data 1.** Raw values for *Figure 2D, F*.
**Figure supplement 1.** Analysis of p53 protein isoform expression in *TP53* dUTR HCT116 cells.

## The endogenous 3′UTR does not regulate p53 translation rates

We wondered whether a potential difference in translation rates of the *TP53* mRNA lacking the 3′UTR might be masked by changes to p53 protein turnover. p53 protein turnover is primarily regulated by the ubiquitin ligase MDM2 (*Momand et al., 1992*; *Haupt et al., 1997*; *Kubbutat et al., 1997*). As transcription of MDM2 itself is activated by p53, the expression of both proteins is balanced by a negative feedback loop that could obscure a change in p53 synthesis rate. In order to assess p53 synthesis in the absence of MDM2-mediated degradation, we treated HCT116 cells with increasing concentrations of the MDM2 inhibitor Nutlin-3. If p53 translation rates from dUTR transcripts were indeed altered, the inhibition of MDM2-mediated p53 degradation would lead to differences in p53 accumulation. However, we observed that the concentration-dependent increase of p53 protein was similar between parental, control clone, and dUTR cells (*Figure 3A and B*).

In addition, we measured de novo p53 protein synthesis directly through metabolic labeling of proteins. For this purpose, we incubated HCT116 cells with the methionine analog azidohomoalanine (AHA) and quantified the relative abundance of labeled p53 protein in steady state and in the presence of etoposide (*Figure 3C*). As expected, etoposide treatment increased the amount of newly synthesisized p53 protein. However, we observed that p53 protein synthesis was not significantly different between cell lines containing or lacking the *TP53* 3′UTR (*Figure 3D and E*). We therefore conclude that the mechanisms involved in regulating p53 protein synthesis and degradation are independent of the 3′UTR.

## The *TP53* coding region has a dominant repressive effect on reporter gene expression and overrides the contribution of the 3′UTR

We next wanted to understand why the removal of the 3′UTR in an endogenous context had such a different effect compared to what was observed in previous studies. Notably, earlier studies that investigated 3′UTR-dependent p53 regulation often used reporter genes as a proxy for endogenous p53 regulation (*Table 1*). We therefore cloned the human *TP53* 3′UTR (1207 nucleotides) or the dUTR fragment (159 nucleotides) downstream of a GFP open reading frame and expressed these constructs in *TP53-/-* HCT116 cells (*Figure 4A*). In the context of this reporter, the full-length *TP53* 3′UTR significantly reduced expression of both GFP mRNA and protein (*Figure 4B and C*) in comparison to the dUTR fragment or the commonly used BGH terminator sequence contained in the vector. This result was recapitulated when luciferase was used instead of GFP as a reporter, thus confirming previous findings (*Figure 4—figure supplement 1A*). We also included the 3′UTRs of three highly expressed human housekeeping genes (*GAPDH*, *HPRT*, and *PGK1*) to evaluate their effect on reporter gene expression. All three yielded higher reporter mRNA and protein expression than the full-length *TP53* 3′UTR but resulted in lower reporter gene expression than the *TP53* dUTR construct, suggesting that the remaining 3′UTR fragment in dUTR cells is unlikely to contain additional repressive elements (*Figure 4C*).

We created a reporter construct called *TP53* 3′UTR (U-del) in which a ~ 40 nucleotide long U-rich element (URE) was deleted which contained the majority of previously annotated binding sites for RNA-binding proteins (*Figure 1A*). Relative to the original full-length 3′UTR, this construct increased

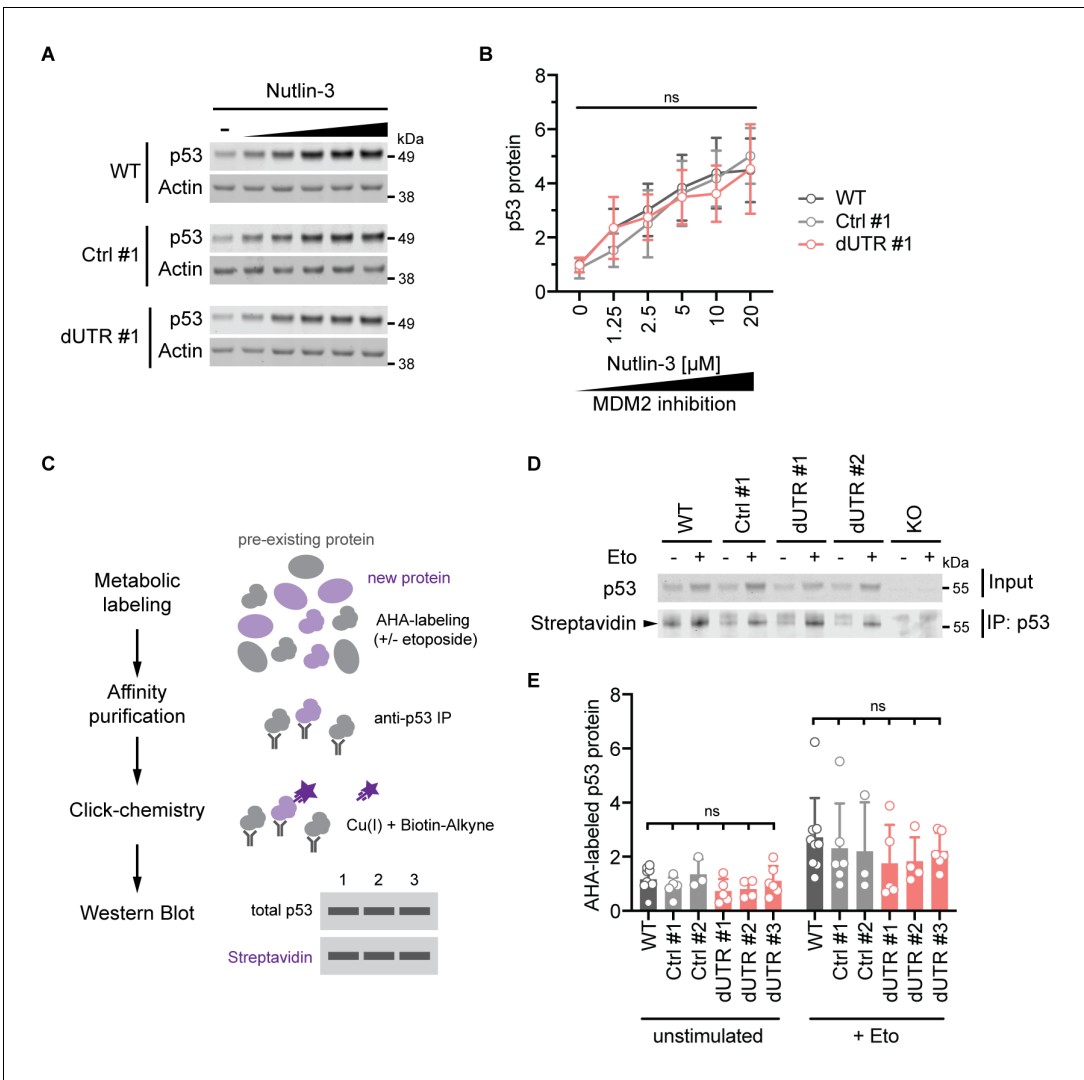

**Figure 3.** The endogenous *TP53* 3'UTR does not impact p53 protein synthesis and turnover. (**A**) Immunoblot showing p53 protein levels after 4 hr of Nutlin-3 treatment (0–20 µM) in WT, Ctrl clone #1, and dUTR clone #1 derived from HCT116 cells. Actin serves as loading control. (**B**) As in (**A**). Quantification of p53 protein expression from n = 4 biological replicates is shown as mean +s.d. Statistical analysis of cell line effect using ANOVA. ns, not significant. See *Figure 3—source data 1* for raw data. (**C**) Experimental setup to analyze p53 protein synthesis by metabolic labeling of proteins using the methionine analog azidohomoalanine (AHA) in the presence or absence of 20 µM etoposide. (**D**) Representative immunoblot for p53 synthesis assessed by metabolic labeling as shown in (**C**). The black triangle indicates the signal specific to p53 protein. (**E**) As in (**D**). Quantification of newly synthesized p53 protein using AHA-labeling for 2 hr. At least three biological replicates were measured in the presence or absence of 20 µM etoposide. Shown is mean +s.d. Statistical analysis using ANOVA. ns, not significant.

The online version of this article includes the following source data for figure 3:

**Source data 1.** Raw values for *Figure 3B*.

---

mRNA and protein expression of the reporter (*Figure 4C*). These data confirm the negative impact of this U-rich element on reporter gene expression.

We next asked whether the endogenous sequence context could explain the apparent discrepancies between reporter assays and endogenous regulation. To address this question, we examined whether different parts of the *TP53* mRNA can impact the expression of our reporter. We added the *TP53* coding region (CDS) to our reporter constructs. We observed that the CDS-GFP fusion protein was expressed at much lower levels than GFP alone (*Figure 4C*, left panel). This could be due, at least in part, to the high p53 turnover stimulated by MDM2. Surprisingly though, we also observed

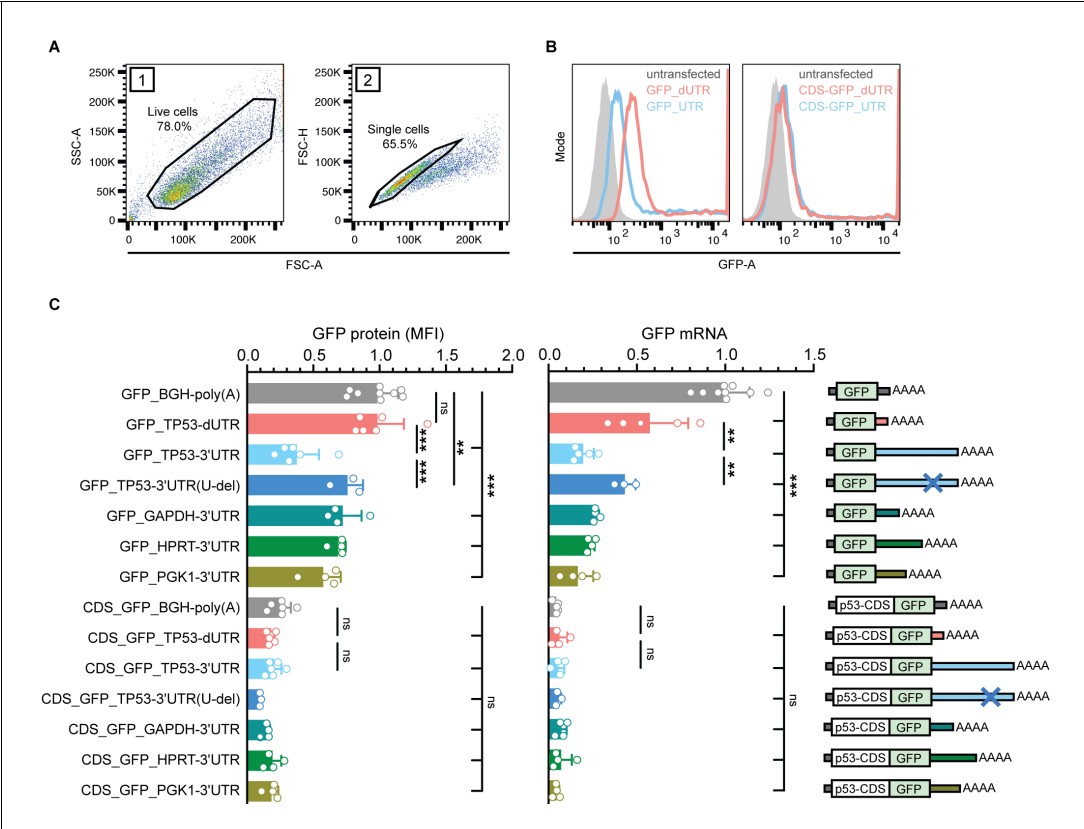

**Figure 4.** The p53 coding region has a dominant repressive effect on the expression of a reporter gene that overrides the contribution of the 3′UTR. (**A**) FACS gating strategy to measure GFP protein expression in *TP53*-/- HCT116 cells. Live cells and single cells were used for downstream analysis. (**B**) Histogram plots from a representative FACS experiment. The gray area represents the untransfected, GFP-negative control population. Shown is GFP fluorescence intensity. (**C**) GFP protein levels were measured as mean fluorescence intensity (MFI) by FACS and GFP mRNA levels were measured by RT-qPCR using *GAPDH* as housekeeping gene in *TP53*-/- HCT116 cells. Shown is mean + s.d. of at least n = 3 independent experiments. CDS, coding sequence. Statistical analysis using ANOVA and Tukey's post-hoc test with \*p<0.05, \*\*p<0.01, \*\*\*p<0.0001; ns, not significant. See ***Figure 4—figure supplement 1*** for additional information.

The online version of this article includes the following figure supplement(s) for figure 4:

**Figure supplement 1.** Validation of repressive effects of the *TP53* 3′UTR in luciferase reporter assays.

significantly reduced levels of p53 CDS-GFP reporter mRNA compared to the GFP reporter. These data suggest a strong contribution of the CDS to *TP53* mRNA stability regulation (***Figure 4C***, right panel). Notably, the relative effect size of the CDS on the mRNA levels of the reporter was larger than with any of the tested 3′UTR constructs. Moreover, in the context of the coding region, none of the tested 3′UTR sequences (including the 3′UTRs of different housekeeping genes) had any impact on mRNA or protein expression of the reporter (***Figure 4C***). Together, these results suggest that the coding region's impact on *TP53* mRNA levels and protein expression is dominant over the 3′UTR and masks the loss of the *TP53* 3′UTR.

Additionally, we asked whether the *TP53* 5′UTR contributes to the regulation of our reporter constructs. Inclusion of the sequence of the endogenous human *TP53* 5′UTR to the reporter constructs that contained the *TP53* CDS did not reveal additional effects on reporter mRNA or protein expression and reporter gene expression remained 3′UTR-independent (***Figure 4—figure supplement 1B***). Together, these results indicate that the *TP53* 3′UTR and CDS functionally interact in the regulation of p53 expression and that the effects of the individual parts are not additive.

# A *Trp53* dUTR mouse model reveals 3′UTR-independent p53 expression in vivo

We reasoned that 3′UTR-dependent p53 expression regulation may still play a role in certain developmental stages, tissues or cell types. In order to explore this possibility, we created an analogous mouse model to investigate the role of the 3′UTR in an organism. We used zygotic injection of a pair of CRISPR/Cas9 guide RNAs to create mice in which we deleted the non-essential part of the mouse *Trp53* 3′UTR (**Figure 5A**). After backcrossing, we analyzed *Trp53* dUTR mice harboring a homozygous 3′UTR deletion (**Figure 5—figure supplement 1A–C**). These mice were viable, fertile, and did not show any developmental defects (**Figure 5—figure supplement 1D and E**). We measured *Trp53* mRNA expression in ten different tissues and did not detect significant differences between samples derived from WT and dUTR mice (**Figure 5B**). To examine the role of the 3′UTR in the regulation of stimulus-dependent p53 expression, we performed total body irradiation of WT and dUTR mice. At

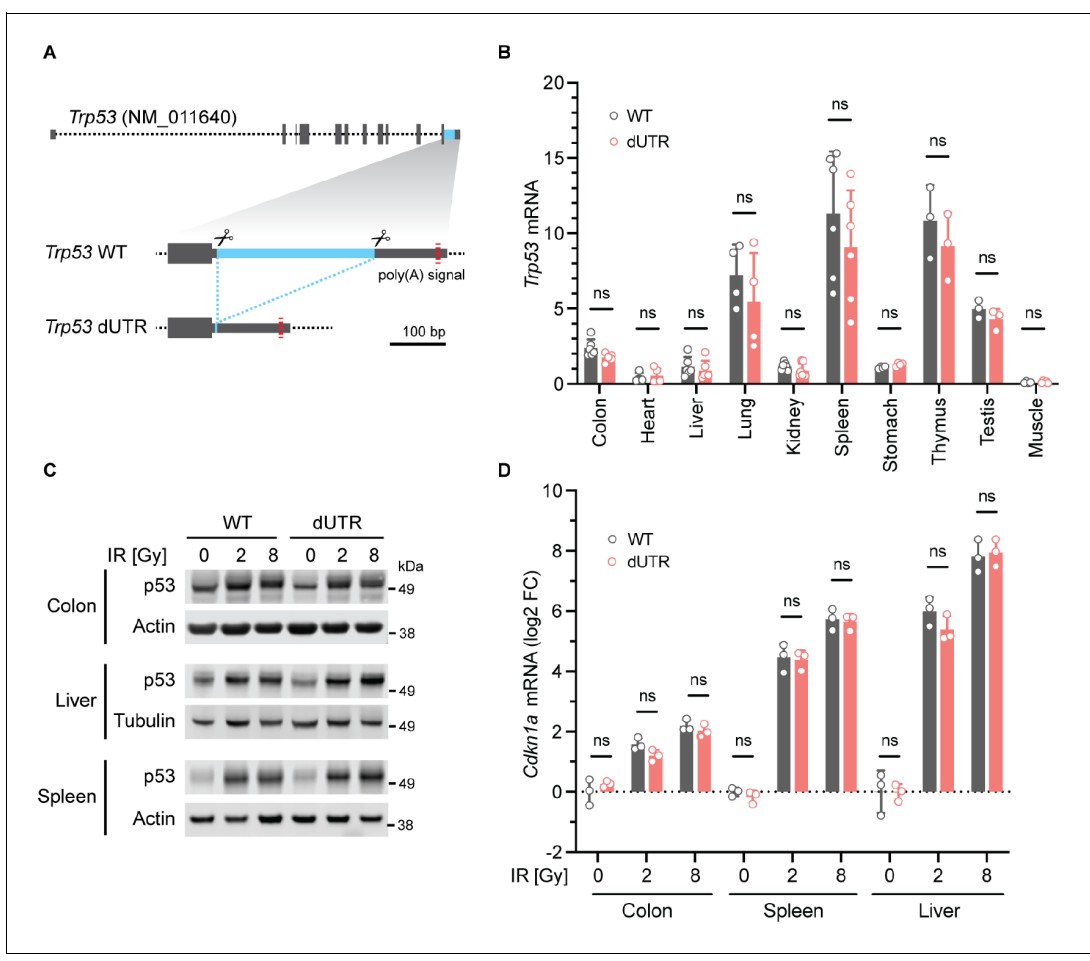

**Figure 5.** Knockout of the *Trp53* 3′UTR does not induce aberrant p53 expression in a mouse model. (**A**) Schematic of the murine *Trp53* gene. The sequence deleted in dUTR cells is shown in blue. (**B**), *Trp53* mRNA in tissues from WT and dUTR mice was normalized to *Gapdh*. Shown is mean + s.d. from n = 3 independent experiments. Statistical analysis using ANOVA. ns, not significant. CRISPR/Cas9-mediated deletions at the nucleotide level are shown in **Figure 5—figure supplement 1C**. (**C**) Representative immunoblots of p53 protein from tissues obtained 4 hr after total body irradiation. Gy, Gray. Tubulin or Actin serve as loading controls. Quantification of p53 protein expression values obtained from n = 3 mice is shown in **Figure 5—figure supplement 2**. (**D**) *Cdkn1a* mRNA expression of samples from (**C**) was normalized to *Gapdh*. Shown is mean + s.d. from three mice. Statistical analysis using ANOVA. ns, not significant.

The online version of this article includes the following figure supplement(s) for figure 5:

**Figure supplement 1.** Generation and characterization of *Trp53* dUTR mice.

**Figure supplement 2.** p53 protein expression in tissues of WT and *Trp53* dUTR mice after whole body irradiation.

4 hr post-irradiation, p53 protein expression was upregulated in spleen, liver, and colon samples to a similar extent in WT and dUTR mice (*Figure 5C*, *Figure 5—figure supplement 2*). We also analyzed expression of *Cdkn1a*, a highly dosage-sensitive p53 target gene that encodes the cell cycle regulator p21 (*el-Deiry et al., 1993*). Four hours after irradiation, *Cdkn1a* mRNA level were equally induced in WT and dUTR mice, suggesting that p53 target gene activation is 3′UTR-independent in mouse tissues (*Figure 5D*). These results demonstrate that the non-essential part of the *Trp53* 3′UTR is not required for steady-state or stimulus-dependent regulation of p53 mRNA or protein level in mice.

## Discussion

The lack of experimental data for 3′UTR-mediated expression regulation in native gene contexts has been a longstanding problem in the field of post-transcriptional gene regulation. Until recently, research on 3′UTR functions has mostly been conducted using overexpression systems and reporter gene assays. In contrast, gene knockouts that disrupt the production of proteins from endogenous gene loci have long been considered the gold standard for analyzing gene functions. The advent of CRISPR/Cas9 gene editing tools has made the generation of 3′UTR knockouts using genomic deletion feasible in both cell lines and organisms.

Using these tools, we conclude that the endogenous *TP53* 3′UTR does not have a significant impact on p53 abundance regulation under standard conditions used to study p53, including induction of DNA damage in HCT116 cells and total body irradiation of mice (*Figures 1D*, *2B, D, H*, *5B and D*). These observations are in stark contrast to the data obtained by overexpression of synthetic constructs in previous studies (*Table 1*). While we could reproduce earlier reporter studies with respect to a repressive function of the *TP53* 3′UTR when used in isolation, we found that the 3′UTR-mediated repressive effect was eliminated in the context of an mRNA containing the *TP53* coding region (*Figure 4C*).

Notably, the *TP53* coding region was the most repressive sequence tested in our reporter system. Our data further support the recently established role of the coding region as a major regulator of mRNA stability and translation (*Presnyak et al., 2015*; *Eraslan et al., 2019*; *Mauger et al., 2019*; *Narula et al., 2019*; *Wu et al., 2019*). Genome-wide comparisons of human coding regions showed that codon optimality and RNA structure in coding regions have the potential to modulate mRNA stability and translation efficiency to a similar extent as 3′UTRs (*Narula et al., 2019*). These effects were often found to be translation-dependent, suggesting the involvement of ribosome-associated protein complexes (*Wu et al., 2019*). The stronger contribution of the coding region is supported by the observation that the coding sequence of *TP53* is more conserved than the 3′UTR sequence. Although this is true for the majority of human genes, the conservation of the *TP53* 3′UTR is among the 20% poorest conserved 3′UTRs of protein coding genes. Given the fatal consequences of p53 deregulation, we speculate that a basic mechanism for p53 expression control may have first evolved in the coding region.

Moreover, we found that the *TP53* coding region abrogated the effect of additional suppressive elements in the 3′UTR. These results corroborated our findings from endogenous p53 regulation, where all sequence elements are located on the same transcript. The impact of sequence context on the effect of *cis*-regulatory elements has been previously described: The 5′UTR, codon optimality and splicing have all been shown to modulate the impact of 3′UTR elements in specific genes (*Cottrell et al., 2017*; *Paolantoni et al., 2018*; *Theil et al., 2018*). However, most studies investigate the effect of untranslated regions in isolation and hence the prevalence of this phenomenon is entirely unknown. Mechanistically, a different sequence context may lead to changes in RNA folding which could create constraints on RNA-binding motif accessibility. In addition, saturation effects and epistatic interactions might limit or modulate the impact of additional suppressive *cis*-regulatory elements provided by the sequence context (*Cottrell et al., 2018*). Our data indicate that the different parts of mRNAs do not act autonomously, but are part of a regulatory unit and functionally cooperate with each other (*Cottrell et al., 2017*; ; *Zrimec et al., 2020*).

3′UTR knockouts can be used to discriminate between direct and indirect targets of miRNAs and RNA-binding proteins. Both miRNAs and RNA-binding proteins usually regulate hundreds of mRNAs. This makes it difficult to assign phenotypes to the deregulation of specific target mRNAs. Moreover, miRNAs and RNA-binding proteins often target several genes in the same pathway (*Ben-*

*Hamo and Efroni, 2015*; *Zanzoni et al., 2019*). Therefore, the results of overexpression or knock-down experiments with miRNAs and RNA-binding proteins may be confounded. This problem has likely contributed to the hypothesis that the 3′UTR is involved in p53 expression regulation. For example, the tumor suppressor RBM38 (also called RNPC1) was proposed to bind to the human *TP53* 3′UTR resulting in lower p53 expression in the presence of RBM38 (*Zhang et al., 2011*). However, apart from *TP53*, RBM38 targets several other genes in the p53 pathway, including *MDM2*, *PPMID*, and *CDKN1A* (*Shu et al., 2006*; *Xu et al., 2013*; *Zhang et al., 2015*). Expression changes of these genes can indirectly upregulate p53 or result in phenotypes that mimic p53 overexpression. Indeed, while RBM38 knockout mice show phenotypes consistent with p53 hyperactivation (*Zhang et al., 2014*), *Trp53* dUTR mice are apparently normal. This suggests that the repressive effects of RBM38 on p53, that were previously attributed to be mediated by direct 3′UTR-dependent expression regulation, may be indirect. Similarly, HUR was shown to bind and regulate expression of both *MDM2* and *CDKN1A*. HUR-deficient mouse embryonic fibroblasts have cell cycle defects that can be rescued by MDM2 overexpression (*Wang et al., 2000*; *Ghosh et al., 2009*).

In addition to the direct regulation of genes in the p53 pathway, we believe it is plausible that some prior studies observed p53 upregulation as a result of general cell stress. Overexpression and knockdown of RNA-binding proteins and miRNAs often result in expression changes across large sets of genes, which could lead to dysfunction in any number of pathways. As an integrator of the cellular stress response, p53 activation is uniquely sensitive to altered cell states. The anti-proliferative transcriptional program that is controlled by p53 is activated upon various stress signals not related to genotoxicity, including ribosomal stress, oxidative stress, loss of adhesion, and oncogene activation (*Horn and Vousden, 2007*). Hence, p53 protein stability may be indirectly regulated by these stress-response pathways as a function of p53 activation. These effects are highly reproducible and extremely difficult to control for.

We want to point out that a recent study that deleted endogenous 3′UTR sequences in several cytokine genes also found discrepancies between reporter-based assays and gene expression from native contexts (*Zhao et al., 2017*). In other cases, endogenous 3′UTR deletions led to the discovery of 3′UTR-dependent mRNA localization phenotypes (*Terenzio et al., 2018*). Although our data indicate that p53 abundance regulation is 3′UTR-independent, the 3′UTR may still have important functions possibly through control of protein localization or protein activity as has been shown for other proteins (*Berkovits and Mayr, 2015*; *Moretti et al., 2015*; *Terenzio et al., 2018*; *Lee and Mayr, 2019*; *Mayr, 2019*; *Bae et al., 2020*; *Fernandes and Buchan, 2020*; *Kwon et al., 2020*). Finally, while our experimental conditions were highly similar to conditions tested in previous studies, we cannot fully exclude the possibility of 3′UTR-dependent expression regulation in select cell types or under specific conditions.

The unexpected finding that the endogenous *TP53* mRNA does not depend on the 3′UTR for post-transcriptional expression regulation as predicted by reporter assays makes a strong case for revising standard experimental procedures. 3′UTR knockouts allow to distinguish between direct and indirect effects associated with post-transcriptional regulation and they enable investigators to perform experiments under endogenous expression conditions. We think that this kind of genetic validation is critical for identifying the most promising gene targets for potential sequence-based miRNA therapeutics that aim to influence gene expression in human patients (*Kasinski and Slack, 2011*; *Hermeking, 2012*; *Bonneau et al., 2019*).

## Materials and methods

**Key resources table**

| Reagent type (species) or resource | Designation | Source or reference | Identifiers | Additional information |
|---|---|---|---|---|
| Strain, strain background (*Mus musculus*) | C57Bl/6J | Jackson Laboratory | #000664 RRID:IMSR_JAX:000664 | used to generate TP53 dUTR mouse strain |

*Continued on next page*

*Continued*

| Reagent type (species) or resource | Designation | Source or reference | Identifiers | Additional information |
|---|---|---|---|---|
| Strain, strain background (*Mus musculus*) | *Trp53* dUTR | This paper | | C57Bl/6J background, see Materials and methods *Supplementary file 1* |
| Cell line (*Homo sapiens*) | FLP In T-REx 293 | From Dr. Thomas Tuschl (Rockefeller University) | RRID:CVCL_U427 | |
| Cell line (*Homo sapiens*) | FLP In T-REx 293 *TP53* dUTR | This paper | | see Materials and methods *Supplementary file 1* |
| Cell line (*Homo sapiens*) | FLP In T-REx 293 *TP53-/-* | This paper | | see Materials and methods *Supplementary file 1* |
| Cell line (*Homo sapiens*) | HCT116 | ATCC | ATCC CCL-247 RRID:CVCL_0291 | |
| Cell line (*Homo sapiens*) | HCT116 Ctrl (two clones) | This paper | | see Materials and methods *Supplementary file 1* |
| Cell line (*Homo sapiens*) | HCT116 *TP53* dUTR (three clones) | This paper | | see Materials and methods *Supplementary file 1* |
| Cell line (*Homo sapiens*) | HCT116 *TP53-/-* | This paper | | see Materials and methods *Supplementary file 1* |
| Peptide, recombinant protein | Cas9 protein with NLS | PNA Bio | CP01-20 | |
| Sequence-based reagent | Costum Alt-R CRISPR Cas9 crRNA (Trp53_gRNA upstream) | IDT | GTGATGGGGACGGGATGCAG | used for CRISPR RNP formation |
| Sequence-based reagent | Costum Alt-R CRISPR Cas9crRNA (Trp53_gRNA downstream) | IDT | CATAGGGTCCATATC CTCCA | used for CRISPR RNP formation |
| Sequence-based reagent | Alt-R CRISPR-Cas9 tracrRNA | IDT | 1072532 | |
| Antibody | Anti-p53 clone DO-7 (mouse monoclonal) | Santa Cruz | sc-47698 RRID:AB_628083 | (1:250) |
| Antibody | Anti-p53 clone PAb240 (mouse monoclonal) | Santa Cruz | sc-99 RRID:AB_628086 | (1:250) |
| Antibody | Anti-p53 clone 1C12 (mouse monoclonal) | Cell Signaling | #2524 RRID:AB_331743 | (1:500) |
| Antibody | Anti-Actin (rabbit polyclonal) | Sigma | A2066 RRID:AB_476693 | (1:1,000) |
| Antibody | Anti-Tubulin (mouse monoclonal) | Sigma | T9026 RRID:AB_477593 | (1:1,000) |
| Antibody | IRDye 800CW anti-Mouse (goat polyclonal) | LI-COR | 926–32210 RRID:AB_621842 | (1:10,000) |
| Antibody | IRDye 680RD anti-Rabbit (goat polyclonal) | LI-COR | 926–68071 RRID:AB_10956166 | (1:10,000) |
| Transfected construct (synthetic) | pX330-U6-Chimeric_BB-CBh-hSpCas9 | Addgene | RRID:Addgene_42230 | |
| Transfected construct (*Homo sapiens*) | pX330-gRNA dUTR1 | This paper | | see Materials and methods *Supplementary file 1* |
| Transfected construct (*Homo sapiens*) | pX330-gRNA dUTR2.1 | This paper | | see Materials and methods *Supplementary file 1* |

*Continued on next page*

*Continued*

| Reagent type (species) or resource | Designation | Source or reference | Identifiers | Additional information |
|---|---|---|---|---|
| Transfected construct (*Homo sapiens*) | pX330-gRNA dUTR2.2 | This paper | | see Materials and methods *Supplementary file 1* |
| Transfected construct (synthetic) | pCDNA3-puro eGFP | PMID:30449617 | | |
| Transfected construct (*Homo sapiens*) | pCDNA3-puro p53(CDS)-eGFP | This paper | | see Materials and methods *Supplementary file 1* |
| Transfected construct (*Homo sapiens*) | pCDNA3-puro eGFP_TP53-3UTR | This paper | | see Materials and methods *Supplementary file 1* |
| Transfected construct (*Homo sapiens*) | pCDNA3-puro p53(CDS)-eGFP_TP53-3UTR | This paper | | see Materials and methods *Supplementary file 1* |
| Transfected construct (*Homo sapiens*) | pCDNA3-puro eGFP_dUTR | This paper | | see Materials and methods *Supplementary file 1* |
| Transfected construct (*Homo sapiens*) | pCDNA3-puro p53 (CDS)-eGFP_dUTR | This paper | | see Materials and methods *Supplementary file 1* |
| Transfected construct (*Homo sapiens*) | pCDNA3-puro eGFP_TP53-3UTR(U-del) | This paper | | see Materials and methods *Supplementary file 1* |
| Transfected construct (*Homo sapiens*) | pCDNA3-puro p53 (CDS)-eGFP_TP53-3UTR(U-del) | This paper | | see Materials and methods *Supplementary file 1* |
| Transfected construct (*Homo sapiens*) | pCDNA3-puro eGFP_GAPDH-3UTR | This paper | | see Materials and methods *Supplementary file 1* |
| Transfected construct (*Homo sapiens*) | pCDNA3-puro p53(CDS)-eGFP_GAPDH-3UTR | This paper | | see Materials and methods *Supplementary file 1* |
| Transfected construct (*Homo sapiens*) | pCDNA3-puro eGFP_HPRT-3UTR | This paper | | see Materials and methods *Supplementary file 1* |
| Transfected construct (*Homo sapiens*) | pCDNA3-puro p53(CDS)-eGFP_HPRT-3UTR | This paper | | see Materials and methods *Supplementary file 1* |
| Transfected construct (*Homo sapiens*) | pCDNA3-puro eGFP_PGK1-3UTR | This paper | | see Materials and methods *Supplementary file 1* |
| Transfected construct (*Homo sapiens*) | pCDNA3-puro p53(CDS)-eGFP_PGK1-3UTR | This paper | | see Materials and methods *Supplementary file 1* |
| Transfected construct (*Homo sapiens*) | pCDNA3-puro 5UTR_p53(CDS)-eGFP | This paper | | see Materials and methods *Supplementary file 1* |
| Transfected construct (*Homo sapiens*) | pCDNA3-puro 5UTR_p53(CDS)-eGFP_dUTR | This paper | | see Materials and methods *Supplementary file 1* |
| Transfected construct (*Homo sapiens*) | pCDNA3-puro 5UTR_p53(CDS)-eGFP_TP53-3UTR | This paper | | see Materials and methods *Supplementary file 1* |

*Continued on next page*

Continued

| Reagent type (species) or resource | Designation | Source or reference | Identifiers | Additional information |
|---|---|---|---|---|
| Transfected construct (synthetic) | psiCHECK-2 | Promega | C8021 | |
| Transfected construct (*Homo sapiens*) | psiCHECK-2_TP53-3UTR | This paper | | see Materials and methods *Supplementary file 1* |
| Transfected construct (*Homo sapiens*) | psiCHECK-2_dUTR | This paper | | see Materials and methods *Supplementary file 1* |
| Chemical compound, drug | IRDye 680LT Streptavidin | LI-COR | 926–68031 | (1:2,000) |
| Chemical compound, drug | Nutlin-3 | Seleckchem | S1061 | |
| Chemical compound, drug | Etoposide | Sigma | 341205–25 MG | |
| Chemical compound, drug | 5-Fluorouracil | Sigma | F6627 | |
| Chemical compound, drug | MTSEA-biotin-XX | Biotium | 900661 | |
| Chemical compound, drug | Biotin Alkyne (PEG4 carboxamide-Propargyl Biotin) | This paper | B10185 | |
| Chemical compound, drug | 4-Thiouridine | MP Biomedicals | MP215213425 | |
| Chemical compound, drug | Yeast tRNA | Invitrogen | 15401029 | |
| Chemical compound, drug | dCTP [$\alpha-32$P] | Perkin Elmer | NEG013H100UC | |
| Commercial assay or kit | Click-iT Protein Reaction Buffer Kit | Invitrogen | C10276 | |
| Commercial assay or kit | Click-iT AHA (L-Azidohomoalanine) | Invitrogen | C10102 | |
| Commercial assay or kit | SuperScript IV Vilo Master Mix | Invitrogen | 11756050 | |
| Commercial assay or kit | Dual-Glo Luciferase Assay System | Promega | E2940 | |
| Commercial assay or kit | Megaprime DNA labeling system, dCTP | Cytiva | RPN1606 | |
| Commercial assay or kit | Lipofectmaine LTX Reagent with PLUS Reagent | Invitrogen | A12621 | |
| Commercial assay or kit | Dynabeads Protein G for Immunoprecipitation | Invitrogen | 10004D | |
| Commercial assay or kit | Dynabeads MyOne Streptavidin C1 | Invitrogen | 65001 | |
| Commercial assay or kit | Oligotex mRNA mini Kit | Quiagen | 70022 | |
| Commercial assay or kit | ULTRAhyb Ultrasensistive Hybridization buffer | Invitrogen | AM8670 | |
| Commercial assay or kit | QuickExtract DNA Extraction Solution | Lucigen | QE09050 | |
| Commercial assay or kit | RNAlater-ICE Frozen Tissue Transition Solution | Invitrogen | AM7030 | |

*Continued on next page*

*Continued*

| Reagent type (species) or resource | Designation | Source or reference | Identifiers | Additional information |
|---|---|---|---|---|
| Commercial assay or kit | SuperScript IV VILO Master Mix with ezDNAse Enzyme | Invitrogen | 11766050 | |
| Commercial assay or kit | FastStart Universal SYBR Green Master (ROX) | Roche/Sigma | 4913850001 | |
| Software, algorithm | FlowJo (Version 10.5.3) | FlowJo, LLC | | |
| Software, algorithm | Prism 8 for OS X (Version 8.4.3) | Graph Pad Software, LLC | | |
| Software, algorithm | Image Studio (Version 5.2) | LI-COR Biosciences | | |

## Generation of the *Trp53* dUTR mouse strain using CRISPR/Cas9

Female C57Bl/6 mice between 3 and 4 weeks of age were superovulated by intraperitoneal injection of Gestyl followed by human chorionic gonadotropin according to standard procedures (*Behringer et al., 2014*). After superovulation, the females were setup with male studs for mating. After mating, fertilized eggs were recovered at the one-cell stage from oviducts of superovulated female mice. One to 2 pl of CRISPR/Cas9 RNP complexes were injected into the pronuclei of fertilized eggs (see details below). Surviving eggs were surgically reimplanted into the oviducts of pseudo-pregnant females previously primed for pregnancy by mating with vasectomized males. The resulting pubs were screened using PCR for the deletion amplicon at 2 weeks of age (primers are listed in *Supplementary file 1*). Suitable candidates were further validated by sequencing.

### Preparation of CRISPR-Cas9 RNP injection mixture

Two target-specific crRNAs and a tracrRNA were purchased from IDT (*Supplementary file 1*). In two separate tubes, 2.5 µg of each crRNA was mixed with 5 µg tracrRNA, heated to 95°C for 5 min and then slowly cooled down to room temperature for annealing. The annealed duplexes were combined and mixed with 1 µg recombinant Cas9 enzyme (PNA Bio) and 625 ng in vitro transcribed Cas9 mRNA and the total volume was adjusted to 50 µl with sterile water.

### Screening for homozygous and heterozygous dUTR mice

Two heterozygous founder males with an identical 295-nucleotide deletion (*Figure 4—figure supplement 1C*) were used to establish a mouse colony. Two or more rounds of backcrossing into wild-type C57Bl/6 mice were performed prior to analysis of *Trp53* dUTR mouse phenotypes. Mouse genotypes from tail biopsies were determined using RT-PCR with specific probes designed for each *Trp53* allele (Transnetyx, Cordova, TN).

### Irradiation of mice

Where indicated, adult mice underwent total body irradiation with 2 or 8 Gy using a Cs-137 source in a Gammacell 40 Exactor (MDS Nordion) at 77 cGy/min. Four hours later irradiated mice were euthanized to collect samples. All procedures were approved by the Institutional Animal Care and Use Committee at MSKCC under protocol 18-07-010.

## Extraction of total RNA from mouse tissues and human cells for RT-qPCR analysis

For RNA extraction from mouse tissue, freshly collected tissue samples were flash-frozen and transferred to RNAlater-ICE Frozen Tissue Transition Solution (Invitrogen). After soaking overnight at −20°C, the tissue samples were homogenized in vials containing 1.4 mm ceramic beads (Fisherbrand) and 400 µl RLT buffer (Qiagen) using a bead mill (Bead Ruptor 24, Biotage). 200 µl of the tissue homogenate was mixed with 1 ml of TRI Reagent (Invitrogen). For extraction of RNA from cultured cells, the cell pellet was directly resuspended in TRI Reagent. Total RNA extraction was

performed according to the manufacturer's protocol. The resulting RNA was treated with 2U DNaseI enzyme (NEB) for 30 min at 37℃, followed by acidic phenol extraction and isopropanol precipitation. To generate cDNA, about 200 ng of RNA was used in a reverse transcription reaction with Super-Script IV VILO Master Mix (Invitrogen). To measure the relative expression levels of mRNAs by RT-qPCR, FastStart Universal SYBR Green Master (ROX) from Roche was used together with gene-specific primers listed in *Supplementary file 1*. GAPDH/Gapdh was used as housekeeping gene.

## Generation of the *TP53* 3′UTR deletion in HCT116 and HEK293 cells

To generate CRISPR/Cas9 constructs, we annealed target-specific DNA sequences and inserted them into a BbsI-digested pX330-U6-Chimeric_BB-CBh-hSpCas9 vector (Addgene plasmid #42230) (*Cong et al., 2013*; *Ran et al., 2013*). One µg of each pX330-gRNA plasmid plus 0.1 µg of pmaxGFP plasmid (Lonza) were transiently transfected into exponentially growing cells using Lipofectamine 2000 (Invitrogen). Three days after transfection, single GFP-positive cells were sorted into 96-well plates and cultured until colonies formed. The genomic DNA from individual cell clones was extracted using QuickExtract DNA Extraction Solution (Lucigen) and screened by PCR for the deletion amplicon using the DNA primers listed in *Supplementary file 1*. In the case of HCT116 cells, we repeated the above-described process using two different heterozygous clones with a new downstream gRNA to obtain homozygous *TP53* dUTR cells. Finally, to validate positive cell clones, all *TP53* alleles of candidate clones were sequenced (*Figure 1—figure supplement 1A*).

## Generation of *TP53* knock-out (KO) HCT116 and HEK293 cells

HCT116 cells were purchased from ATCC and Flp-In T-REx 293 cells were a gift from the lab of Thomas Tuschl (Rockefeller University). The cell lines were not authenticated. They are free of mycoplasma. Mycoplasma detection was performed by DAPI staining. We generated our own *TP53*-deficient HEK293 and HCT116 cell lines by targeting exon 6 of the *TP53* coding region with a gRNA causing frame shift mutations. Specifically, pX330 plasmid harboring a *TP53*-specific gRNA (*Supplementary file 1*) was transfected into HEK293 and HCT116 cells using Lipofectamine 2000 (Invitrogen). Two days later, the cells were split and seeded sparsely on a 10 cm dish in the presence of 10 µM Nutlin-3 (Seleckchem), which was used to select against growth of p53-competent cells. After 10 days, single colonies were picked, and individual clones were validated by western blot for loss of p53 expression.

## Western blot analysis

RIPA buffer (10 mM Tris-HCL pH 7.5, 150 mM NaCl, 0.5 mM EDTA, 0.1% SDS, 1% Triton X-100, 1% deoxycholate, Halt Protease Inhibitor Cocktail [Thermo Scientific]) was used to extract total protein from cultured cells or mouse tissues. Cell pellets were washed with PBS and directly resuspended in lysis buffer and incubated on ice for 30 min. Mouse tissue samples were homogenized in RIPA buffer using a bead mill in vials filled with 1.4 mm ceramic beads. Tissue lysates were sonicated to shear genomic DNA prior to removing insoluble components by centrifugation (10 min, 15,000 g). The proteins in the supernatant were precipitated by adding 0.11 volumes of ice-cold 100% Trichloroacetic acid (TCA) and incubated at −20℃ for 1 hr. The samples were centrifuged (10 min, 15,000 g) and the pellet was washed twice in ice-cold acetone before resuspending in reducing 2x Laemmli buffer (Alfa Aesar). Proteins were separated by size on a 4–12% Bis-Tris SDS-PAGE gels (Invitrogen) and blotted on a 0.2 µm nitrocellulose membrane (Bio-Rad). The membrane was then incubated with primary antibody in Odyssey Blocking buffer (LI-COR) overnight at 4℃. The following primary antibodies were used in this study: anti-human p53 DO-7 (Santa Cruz, sc-47698, mouse, 1:250), anti-human p53 PAb240 (Santa Cruz, sc-99, mouse, 1:250), anti-mouse p53 1C12 (Cell Signaling, #2524, mouse, 1:500), anti-Actin (Sigma, A2008, rabbit, 1:1000), anti-Tubulin (Sigma, T9026, mouse 1:1000), and anti-GAPDH (Sigma, G8705, mouse, 1:1000). After washing, the membrane was incubated with fluorescently-labeled secondary antibodies (IRDye 800CW Goat anti-Mouse, 926–32210; IRDye 680LT Goat anti-Rabbit, 926–68071 LI-COR; IRDye 680LT Streptavidin 926–68031) and signals were recorded using the Odyssey Infrared Imaging system (LI-COR). Signals were quantified using the Image Studio 5.2 software.

## Northern blot

Total RNA from cells was extracted as described above. Afterwards, polyA+ mRNA was enriched from total RNA using the Oligotex suspension (Qiagen) according to the manufacturer's instructions. 1.2 µg of polyA+ mRNA was glyoxylated and run on an agarose gel as described previously (*Mayr and Bartel, 2009*, dx.doi.org/10.17504/protocols.io.bqqymvxw). The RNA was transferred overnight using the Nytran SuPerCharge TurboBlotter system (Whatman) and UV-crosslinked.

DNA probes complementary to the *TP53* coding region or the 3′UTR were labeled with dCTP [α-$^{32}$P] using the Megaprime DNA labeling system (Cytiva). Primers used for probe synthesis from human cDNA are listed in *Supplementary file 1*. Labeled probes were denatured by heat for 5 min at 90°C and then incubated with the blot in ULTRAhyp Ultrasensitive Hybridization Buffer (Invitrogen) overnight at 42°C. The blot was washed three times and exposed on a phosphorimaging screen. The radioactive signal was acquired using the Fujifilm FLA700 phosphorimager.

## Human cell culture and drug treatment

Human cell cultures were maintained in a 5% $CO_2$/37°C humidified environment. HEK293 cells were cultured in DMEM (high glucose) and HCT116 cells were cultured in McCoy's 5A medium which were supplemented with 10% FBS and 1% Penicillin/Streptomycin. Where indicated, HCT116 cells were treated with etoposide (0.125–32 µM, Sigma), 5-fluorouracil (40 µM, Sigma), Nutlin-3 (1.25–20 µM, Seleckchem), or UV (50 J/m$^2$) prior to downstream analysis.

## Measurement of mRNA half-life

*TP53* mRNA half-lives were estimated following the protocol described by *Russo et al., 2017* with minor modifications. Briefly, HCT116 cells were treated with 250 µM 4-thiouridine (4sU, MP Biomedicals) in the presence or absence of 20 µM etoposide for four hours. Additionally, reference samples with saturated 4sU-labeled RNA were generated by incubating HCT116 cells at a concentration of 100 µM 4sU for 3 days. After labeling, TRI Reagent (Invitrogen) was used to extract total RNA from cells. In the subsequent biotinylation reaction 40 µg of total RNA was mixed with 0.4 µg unlabeled yeast RNA, 10x biotinylation buffer (100 mM HEPES pH 7.5, 10 mM EDTA) and 10 µg MTSEA-biotin-XX (Biotium) in a total volume of 150 µl. The reaction was incubated for 30 min at room temperature and the RNA was recovered by ethanol precipitation. Then, half of the biotinylated RNA was mixed with 100 µM DTT and retained as the total RNA fraction. The other half was incubated with Dynabeads MyOne Streptavidin C1 magnetic beads (Invitrogen, blocked with 1% BSA and 1 µg/ml yeast tRNA) for 20 min at room temperature. Subsequently, the beads were washed three times with high-salt buffer (100 µM Tris-HCl pH 7.5, 1M NaCl and 0.1% NP40) using a magnet. The bead-bound, newly synthesized RNA fraction was eluted twice in 100 µM DTT and the resulting pooled RNA fraction was precipitated with ethanol. Total and newly synthesized RNA were reverse transcribed using the SuperScript VILO master mix kit. Relative enrichment of *TP53* mRNA in newly synthesized over total fractions was determined by RT-qPCR relative to two yeast mRNAs (see *Supplementary file 1* for primer sequences). Reference values obtained from fully labeled RNA samples were used to estimate the fraction of labeled *TP53* mRNA (R) in each experiment. mRNA half-lives were then calculated according to the following formula:

$$t_{\frac{1}{2}} = -(4h) * \frac{\ln(2)}{\ln(1-R)}$$

$$R = \frac{\frac{newly\ synthesized}{total}\ RNA\ (sample)}{\frac{newly\ synthesized}{total}\ RNA(reference)}$$

## Detection of newly synthesized p53 protein by metabolic labeling

HCT116 cells were grown in a 10 cm dish until 70–80% confluency. Cells were washed with PBS, and methionine-free McCoys medium was added for one hour to deplete cellular methionine reserves. Afterwards, Click-IT AHA (Invitrogen) was added to a final concentration of 50 µM and incubated for two hours at 37°C. Where indicated, 20 µM etoposide was added together with AHA.

Cells were trypsinized and cell pellets were resuspended in 1 ml lysis buffer (10 mM Tris-HCL pH 7.5, 150 mM NaCl, 0.5% NP-40, and protease inhibitors). After brief sonication, the cleared lysate

was mixed with p53 antibody DO-7 (Santa Cruz, sc-47698) at a concentration of 1 µg/ml and incubated for 4 hr at 4℃ while rotating. Equilibrated Dynabeads Protein G (Invitrogen) were added to each lysate and incubated for another hour at 4℃. The beads were separated by a magnet and washed three times with lysis buffer. Bead-bound, AHA-labeled p53 protein was biotinylated using the Click-IT AHA protein labeling kit (Invitrogen) according to the manufacturer's instructions. Free biotin was removed by washing the beads twice in PBS using a magnet. The beads were resuspended in Laemmli SDS sample buffer and heated to 95℃ for 5 min. The samples were loaded on a 4–12% NuPAGE Bis-Tris gel (Invitrogen) and blotted on a nitrocellulose membrane (Bio-Rad). The blots were stained with IRDye680LT Streptavidin (LI-COR) and signals were recorded with the Odyssey Infrared Imager (LI-COR).

## Reporter assays

### Constructs

We PCR-amplified the *TP53* 3′UTR sequence (nucleotides 1380 to 2586 of the reference mRNA NM_000546, May 2018) from WT HCT116 cDNA. This sequence was cloned downstream of the stop codon in pcDNA3.1-puro-eGFP using EcoRI/NotI restriction enzymes. For the dUTR construct, cDNA from *TP53* dUTR HCT116 cells was used to amplify the remaining 3′UTR sequence after CRISPR-mediated deletion, representing a fusion of the first 12 and the last 147 nucleotides of the full-length *TP53* 3′UTR. A *TP53* 3′UTR construct with a deletion of a poly-U stretch (deletion of nucleotides 2123 to 2162 of the reference mRNA NM_000546, May 2018) was generated by overlap extension PCR with the primers listed in *Supplementary file 1*. The *TP53* coding region, encoding the α protein isoform (1182 nucleotides), was cloned upstream and in- frame of the GFP-cassette using HindIII/BamHI restriction sites. The 3′UTRs of human *GAPDH*, *HPRT* and *PGK1* housekeeping genes were PCR-amplified from a cDNA library with primers listed in *Supplementary file 1* and inserted into the vector using EcoRV/NotI restriction sites. The 5′UTR of human *TP53* (nucleotides 1 to 202 of reference mRNA NM_000546, May 2018) was synthesized as a Geneblock from Genewiz and inserted into the vector upstream of the *TP53* coding region using NheI/HindIII restriction sites. For luciferase reporter studies, the full-length 3′UTR and dUTR sequences described above were cloned into a SmaI-digested psiCHECK2 (Promega) vector via blunt-end cloning.

### GFP reporter

*TP53-/-* HCT116 cells were transfected with equimolar amounts of GFP-containing reporter constructs using Lipofectamine LTX Reagent (Invitrogen) according to the manufacturer's instructions. 24 hr after transfection, cells were harvested for analysis of GFP mRNA and protein expression. GFP protein levels were analyzed by flow cytometry using a BD LSRFortessa Flow Cytometer. Raw data were analyzed using the FlowJo software package and mean fluorescence intensity (MFI) values of live single cells were normalized to values obtained from GFP-BGH poly(A) constructs. mRNA abundance of the GFP reporter was measured using RT-qPCR as described above using primers listed in *Supplementary file 1*. The GFP reporter mRNA was normalized to *GAPDH* mRNA.

### Luciferase reporter assay

Luciferase activity was measured 24 hr after transfection of equimolar amounts of psiCHECK2 plasmids (Promega) containing the *TP53* 3′UTR, the dUTR sequence or no insert downstream of the Renilla luciferase translational stop codon. Cells were lysed in passive lysis buffer and Renilla and firefly luciferase activity were measured in duplicates using the Dual-Glo Luciferase Assay System (Promega) according to the manufacturer's instructions in a GloMax 96 Microplate Luminometer (Promega). Relative light units of Renilla luciferase were normalized to firefly luciferase activity.

## Conservation analysis

UCSC phastCons conservation scores for the human genome (hg19) were calculated from multiple alignments with 99 vertebrate species and the mean value for each 3′UTR was obtained. The conservation analysis of binding sites in the 3′UTR sequence as shown in *Table 1* was performed by averaging the phyloP scores of the corresponding genomic regions (100way for human and 35way for mouse).

## Statistics and reproducibility

Statistical analysis of the mRNA and protein expression data was performed using a Student's t-test or ANOVA followed by a Tukey's multiple comparison test. We use ns ($p>0.05$), * $0.01 < p < 0.05$, ** $0.001 < p < 0.01$, and ***$p<0.001$ to indicate the levels of p-values in figures. No data were excluded. The results for immunoblotting are representative of at least three biologically independent experiments. All statistical analyses and visualizations were performed using GraphPad (Prism 8).

## Acknowledgements

We thank all members of the Mayr lab for helpful discussions and critical reading of the manuscript. We thank the Mouse Genetics Core Facility at MSKCC for assistance in the generation of *Trp53* dUTR mice. This work was funded by a postdoctoral fellowship from the DFG to SM and by the NIH Director's Pioneer Award (DP1-GM123454), the Pershing Square Sohn Cancer Research Alliance to CM, and the NCI Cancer Center Support Grant (P30 CA008748). The funders had no role in study design, data collection and interpretation, or the decision to submit the work for publication.

## Additional information

### Funding

| Funder | Grant reference number | Author |
|---|---|---|
| NIH Office of the Director | DP1-GM123454 | Christine Mayr |
| Pershing Square Foundation | | Christine Mayr |
| National Cancer Institute | P30 CA008748 | Christine Mayr |
| Deutsche Forschungsgemeinschaft | | Sibylle Mitschka |

The funders had no role in study design, data collection and interpretation, or the decision to submit the work for publication.

### Author contributions

Sibylle Mitschka, Conceptualization, Formal analysis, Validation, Investigation, Visualization, Methodology, Writing - original draft, Writing - review and editing; Christine Mayr, Conceptualization, Supervision, Funding acquisition, Writing - original draft, Writing - review and editing

### Author ORCIDs

Christine Mayr (iD) https://orcid.org/0000-0002-7084-7608

### Ethics

Animal experimentation: This study was performed in strict accordance with the recommendations in the Guide for the Care and Use of Laboratory Animals of the National Institutes of Health. All of the animals were handled according to approved institutional animal care and use committee (IACUC) protocols (#18-07-010) of Memorial Sloan Kettering Cancer Center. All procedures were approved by the Institutional Animal Care and Use Committee at MSKCC under protocol #18-07-010.

### Decision letter and Author response

Decision letter https://doi.org/10.7554/eLife.65700.sa1
Author response https://doi.org/10.7554/eLife.65700.sa2

# Additional files

## Supplementary files
- Supplementary file 1. Primer sequences.
- Transparent reporting form

## Data availability
All data generated or analyzed during this study are included in the manuscript and supporting files.

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
