## [Decision Letter]

**Acceptance summary:**

Through a series of elegant data, this study is the first to show that the 3'UTR of p53 mRNA is not important for p53 expression. This work emphasizes the need for genetic approaches to study post-transcriptional gene regulation.

**Decision letter after peer review:**

Thank you for sending your article entitled "Endogenous p53 expression in human and mouse is not regulated by its 3′UTR" for peer review at *eLife*. Your article has been reviewed by 3 peer reviewers, one of whom is a member of our Board of Reviewing Editors, and the evaluation has been overseen by Maureen Murphy as the Senior Editor.

Although the Reviewers all acknowledged the novelty of your system and the rigor of the findings presented, all three felt that certain key experiments were missing, particularly regarding RNA half-life and translation, in both stressed and unstressed cells. After conferring, the reviewers felt that the amount of new work requested was likely to be considerable, and potentially prohibitive.

*Reviewer #1:*

In this manuscript the authors investigated the role of the 3'UTR of p53 mRNA on endogenous p53 expression in human and mouse cells. The authors utilized CRISPR/Cas9 to delete the p53 3'UTR and found that deletion of the 3'UTR does not significantly affect the expression of the p53 protein and that of p21 mRNA, a well-established transcriptional target of p53. Although the data presented are interesting and contradicts several previous reports from different labs where the p53 3'UTR was shown to be regulated by miRNAs and RNA-binding proteins such as HuR, there are many concerns that should be addressed before further consideration. Indeed, even if the authors find that the 3UTR of p53 is not important for basal p53 expression and p53 induction in response to the DNA damaging agent, from a broader perspective, the significance of these findings is unclear to this reviewer.

Specific comments that need to be addressed are:

1. The authors should specifically describe how portions of the p53 3'UTR were determined to be essential or non-essential when determining the region to delete.

2. P53 is known to play an important role in G1 arrest in response to DNA damage. For example, in HCT116 cells, loss of p53 results in almost no G1 arrest upon Doxorubicin treatment. The authors should determine if there are differences in cell cycle distribution between the WT, dUTR and p53KO cells upon Doxorubicin treatment (300 nM, 48 to 72 hr of continuous treatment).

3. In Figure 3c, it is more beneficial to clone both the p53 coding sequence and the 3'UTR or dUTR fragment downstream of GFP, rather than having the coding sequence upstream and the 3'UTR downstream. Having the 3'UTR proximal to the coding sequence is more representative of biological conditions.

4. It appears that the WT 293T or HCT116 cells are parental cells. This may not be a rigorous control because these cells were not transfected and did not go through the process of clonal selection during CRISPR/Cas9. The ideal control for the dUTR cell lines would be to use clonal cell lines generated in parallel to the dUTR cells using a non-targeting gRNA or a wild-type clone. All Western blots and RT-qPCR experiments should be repeated using one of these proper negative controls.

5. p53 is regulated predominantly at the level of protein stability. Because miRNAs or RNA-binding proteins might be involved in fine-tuning p53 expression, the role of the 3'UTR might be underestimated if the immunoblots were not quantitated. The authors to provide quantifications of the immunoblot images, assuming they can use a detection system in the linear range.

6. It is important to determine if deletion of the 3'UTR results in compensatory mechanisms to restore p53 expression. To test this, the authors should compare the stability of endogenous p53 protein between the WT clones and dUTR clones in HCT116 and HEK293T cells. Additionally, by S35 methionine pulse chase experiments, the authors should compare the rate of translation of endogenous p53 in HCT116 and HEK293T WT vs dUTR clones. Is it possible that deletion of the 3'UTR results in decreased translation and increased p53 protein stability to restore p53 expression? The experiments to address this question should be repeated at least 3 times to get statistically relevant data.

7. The TP53 gene also expresses other p53 isoforms and an antisense transcript called Wrap53. Does deletion of the 3'UTR have any effect on the p53 isoforms and the antisense lncRNA? Addressing this could be important because loss of the p53 3UTR can affect the expression of these genes and profoundly alter the response of cells to DNA damaging agents.

8. The authors claim that others have shown regulation of p53 3UTR using reporter assays only. However, endogenous p53 mRNA translation was shown to be regulated by HuR. PMID: 14517280, 12821781.

9. In figure 1a, within the 3'UTR there is a particular segment in which binding sites for multiple RNA-binding proteins overlap. Is there any effect of deleting only this segment?

*Reviewer #2:*

In this interesting manuscript "Endogenous p53 expression in human and mouse is not regulated by its 3′UTR" Mitschka and Mayr examine the gene-regulatory potential of the evolutionarily conserved 3'untranslated region (UTR) of the human and mouse p53 gene (TP53). Using CRISPR-Cas9-mediated gene editing they carefully deleted the entire 3'UTR in human cell lines and mice, leaving processing elements intact. They show that deleting the potential regulatory elements in the TP53 3'UTR does not influence steady state p53 protein expression, as well as p53 protein activation after genotoxic stress.

The authors strongly imply that the TP53 3'UTR does not play an important role in TP53 gene function, nevertheless, the conservation of recognizable regulatory elements, such as miRNA binding sites is difficult to square with a lack of function. It would seem necessary to adress this seeming contradiction constructively, considering that it appears to undercut the importance of untranslated regions – if the conserved UTR of a key gene, such as TP53 does not convey some function, are UTRs even important?

The presented experiments are unobjectionable and performed to a high standard. That being said, the author's assays, particularly the experiments relating to p53 activation under genotoxic stress mainly cover two aspects: a) translational regulation of the TP53 mRNA and b) posttranslational activation of p53 protein. Considering that the vast majority of the literature points to p53 being activated by genotoxic stress by increasing p53 protein halflife, mainly due to prevention of p53 ubiquitination by MDM2, it is not particularly surprising that once steady-state levels of p53 protein are set, the 3'UTR itself should not play any important role. It is furthermore not surprising that expression of their reporter with the GFP-p53CDS fusion is efficiently suppressed, given that its levels will be controlled by MDM2 in a posttranslational manner. What would happen if steady-state levels of p53 are manipulated, e.g. by simultaneous depletion of MDM2?

With p53 being a key protein for cell survival, it is conceivable that its levels are tightly controlled, possibly by multiple buffer systems, not just the UTR alone. E.g. the authors never test whether the half-life of the mRNA itself is impacted – in Figure 1c, they show a trend towards downregulation of steady-state mRNA levels; this in turn could indicate that either transcriptional activity or mRNA half-life are affected. Putting numbers on these parameters might shed light on the presence of additional buffer systems in the cell/mouse ensuring proper p53 protein levels in a sort of fail-safe mechanism.

Finally, the authors do not explore the possibility that the effects of their knockouts can be buffered by compensatory mechanisms (e.g. described by the Stainier group that shows that for some genes acute loss by shRNA/siRNAs shows different and more severe phenotypes than gene editing). Of course, this is difficult to test; possibly rescue of full TP53 KO with constructs containing or lacking the TP53 UTR could shed some light on the mRNA dynamics.

In summary, by heavily relying on their genotoxic assays that are likely to depend on steady-state p53 protein levels, the authors may have been looking in the wrong place to observe an effect of the TP53 UTR on mRNA dynamics. While their main observation that TP53 UTR deletion by itself does not have a clear phenotype, and miRNA based therapies explored in the context of p53 are likely to fail, they miss the opportunity to address the question whether the TP53 UTR impacts mRNA dynamics and whether the presence of additional buffering mechanisms obscures a direct phenotype.

Three major points to be considered:

– mRNA halflife and transcription: Putting numbers on these parameters might shed light on the presence of additional buffer systems in the cell/mouse ensuring proper p53 protein levels in a sort of fail-safe mechanism.

– Acute vs chronic loss of p53 UTR: possibly rescue of full TP53 KO with constructs containing or lacking the TP53 UTR could shed some light on the mRNA dynamics.

– Considering that once p53 steady-state levels are set by the activity of MDM2, the effect of genotoxic stress may be predictable: What would happen if steady-state levels of p53 are manipulated, e.g. by simultaneous depletion of MDM2?

*Reviewer #3:*

By utilizing the CRISPR approach, authors specifically delete the majority of the 3'UTR from the endogenous p53 gene in multiple human cell lines as well in mouse. Their study revealed that deletion of the 3'UTR does not alter the mRNA as well as protein levels of p53 under normal as well as DNA damage. Based on these studies, the authors propose that unlike the findings of others in the literature, the 3'UTR does not seem to play crucial roles in regulating p53 mRNA levels or translation of the endogenous TP53 gene.

In general, this is a nicely articulated manuscript with interesting data. The data argue against the role of 3'UTR in the differential expression of p53. However the significance of the findings are not entirely clear to this reviewer, and it is felt that the authors may need to perform experiments to test the role of other elements (5'UTR as well as coding region) in regulating p53 stability as well as translation.

1. The steady state levels of a protein is dictated by regulated translation versus protein degradation, and each of these processes are controlled by distinct sequence elements located within the transcript. Does the 3'UTR of p53 influence differential translation? The authors really should check the polysome association of WT as well dUTR p53 mRNA.

2. Both the WT and the 3'UTR present in the dUTR constructs contain a small region, which based on earlier studies, can support the binding of a few RBPs and miRNAs. It cannot be completely ruled out that those interactions might influence the post transcriptional processing of the p53 mRNA. It might be worth doing an experiment in which the entire 3'UTR of the endogenous p53 could be replaced with the 3'UTR of a house-keeping gene to see the effect on p53 mRNA stability, polysome association and translation.

3. The reporter assays indicated that both the 3'UTR as well as the CDS region contribute to the steady state levels of p53 mRNA. It is not clear why the dUTR alone containing reporter showed increased steady state levels of reporter mRNA compared to WT, if the 3'UTR of p53 does not have any function in regulating the stability of p53 mRNA. This needs to be better-explained.

---

## [Author Response]

Although the Reviewers all acknowledged the novelty of your system and the rigor of the findings presented, all three felt that certain key experiments were missing, particularly regarding RNA half-life and translation, in both stressed and unstressed cells. After conferring, the reviewers felt that the amount of new work requested was likely to be considerable, and potentially prohibitive.Reviewer #1:In this manuscript the authors investigated the role of the 3'UTR of p53 mRNA on endogenous p53 expression in human and mouse cells. The authors utilized CRISPR/Cas9 to delete the p53 3'UTR and found that deletion of the 3'UTR does not significantly affect the expression of the p53 protein and that of p21 mRNA, a well-established transcriptional target of p53. Although the data presented are interesting and contradicts several previous reports from different labs where the p53 3'UTR was shown to be regulated by miRNAs and RNA-binding proteins such as HuR, there are many concerns that should be addressed before further consideration. Indeed, even if the authors find that the 3UTR of p53 is not important for basal p53 expression and p53 induction in response to the DNA damaging agent, from a broader perspective, the significance of these findings is unclear to this reviewer.

We thank the reviewer for spending time to review our manuscript and for the insightful comments. In the revised manuscript, we added additional genetic controls, investigated potential compensatory mechanisms, and refined our data interpretation.

Specific comments that need to be addressed are:1. The authors should specifically describe how portions of the p53 3'UTR were determined to be essential or non-essential when determining the region to delete.

3′UTRs are known to be involved in two important processes: 1) Co-transcriptional 3′ end processing and 2) post-transcriptional gene regulation. The first of these functions is essential for gene expression, as a failure to cleave and polyadenylate a transcript will result in decreased or abrogated expression of the mRNA and protein. Our aim was to design the 3′UTR deletion so that the remaining sequence still allows the endogenous cleavage and polyadenylation machinery to efficiently process the nascent *TP53* transcript. To make an informed decision about the location of sequences involved in 3′ end processing, we re-analyzed ChIP-seq data for all relevant 3′ end processing factors on a genome-wide level (Martin et al., 2012). In agreement with prior analyses, we determined that the binding sites of core factors as well as auxiliary elements are usually confined to a region encompassing the last 100-150 nucleotides upstream of the cleavage site (see Author response image 1). Therefore, we consider the 150 nucleotides located upstream of the cleavage site as essential.

Of note, the read coverage from CLIP-seq data for binding sites of individual cleavage factors were too sparse to map them precisely to the *TP53* mRNA. However, using global 3′ end processing factor binding patterns and established criteria for guide RNA selection, we designed a deletion that left only 147 nucleotides of the original 3′UTR upstream of the cleavage site. This rationale is described in the text (lines 82-93).

**Author response image 1. sa2fig1:** mRNA cleavage and polyadenylation requires multiple sequence elements surrounding the cleavage site. A, Schematic of the multiprotein complex responsible for mRNA cleavage and polyadenylation in humans. B, Sequence context of functional poly(A) sites showing the poly(A) signal hexamer as well as two common auxiliary motifs. The metaplot is aligned to the transcript end in the longest isoform of RefSeq-annotated human genes. C-E, Metagene analysis shows densities of the binding sites of protein components of the cleavage and polyadenylation machinery determined by CLIP: C, Cleavage Factor I complex D, Cleavage and Polyadenylation Specificity Factor complex and E, Cleavage Stimulation Factor complex. Binding sites were retrieved from the POSTAR2 database and aligned to RefSeq (Y. Zhu et al., 2019).

2. P53 is known to play an important role in G1 arrest in response to DNA damage. For example, in HCT116 cells, loss of p53 results in almost no G1 arrest upon Doxorubicin treatment. The authors should determine if there are differences in cell cycle distribution between the WT, dUTR and p53KO cells upon Doxorubicin treatment (300 nM, 48 to 72 hr of continuous treatment).

In this manuscript, we focused on the influence of the *TP53* 3′UTR on p53 abundance regulation. In our opinion a thorough investigation of potential p53 functions is beyond the scope of this paper. However, as suggested by the reviewers, we have performed additional experiments that allowed us to measure mRNA and protein synthesis which further strengthen our claims on p53 abundance regulation (Figure 1E, 1F and new Figure 3A-E). Furthermore, we added a sentence to the ‘Discussion’ where we state the limitations of our work. Although we investigated 3′UTR-dependent p53 expression regulation using the most frequently used stress conditions, including DNA damage, UV, and γ-irradiation (line 322-324), we cannot entirely rule out that a condition exists where the p53 3′UTR regulates p53 abundance.

3. In Figure 3c, it is more beneficial to clone both the p53 coding sequence and the 3'UTR or dUTR fragment downstream of GFP, rather than having the coding sequence upstream and the 3'UTR downstream. Having the 3'UTR proximal to the coding sequence is more representative of biological conditions.

We had the same thought and initially designed constructs for N-terminally tagged p53. However, we found that N-terminally tagged p53 had an unusual subcellular localization (data not shown). Thus, in line with most p53 studies, we decided to use C-terminally tagged p53 constructs which would ensure the expression of a functional p53 protein as shown by others (Norris et al. 1997, Stewart-Ornstein et al. 2017). The reporter data are now presented in new Figure 4 (formerly Figure 3).

4. It appears that the WT 293T or HCT116 cells are parental cells. This may not be a rigorous control because these cells were not transfected and did not go through the process of clonal selection during CRISPR/Cas9. The ideal control for the dUTR cell lines would be to use clonal cell lines generated in parallel to the dUTR cells using a non-targeting gRNA or a wild-type clone. All Western blots and RT-qPCR experiments should be repeated using one of these proper negative controls.

We agree with the reviewer and have added two control clones that underwent the genome editing procedure but have intact 3′UTRs (see Figure 1-Figure supplement 1A) to all experiments presented in the revised manuscript, including RT-qPCRs, western blots, mRNA half-life analysis, and p53 protein production. Of note, we indeed saw a difference between all clones and the parental HCT116 cell line in *TP53* mRNA half-life, however the difference was not associated with the dUTR genotype (Figure 1E, 1F).

5. p53 is regulated predominantly at the level of protein stability. Because miRNAs or RNA-binding proteins might be involved in fine-tuning p53 expression, the role of the 3'UTR might be underestimated if the immunoblots were not quantitated. The authors to provide quantifications of the immunoblot images, assuming they can use a detection system in the linear range.

We used an NIR fluorescence-based detection system (LI-COR) for recording western blot results which enables linear signal detection over a large signal range. Thus, we were able to provide quantifications for all prior and new western blot experiments in the revised version of the manuscript (see new Figure 1-Figure supplement 1D, Figure 2B, 2D, 2F, 2H, 3B, 3E and Figure 5-Figure supplement 2). Each experiment was performed independently at least three times and we used statistics to evaluate significance.

6. It is important to determine if deletion of the 3'UTR results in compensatory mechanisms to restore p53 expression. To test this, the authors should compare the stability of endogenous p53 protein between the WT clones and dUTR clones in HCT116 and HEK293T cells. Additionally, by S35 methionine pulse chase experiments, the authors should compare the rate of translation of endogenous p53 in HCT116 and HEK293T WT vs dUTR clones. Is it possible that deletion of the 3'UTR results in decreased translation and increased p53 protein stability to restore p53 expression? The experiments to address this question should be repeated at least 3 times to get statistically relevant data.

We acknowledge the possibility of compensation through feedback regulation. We measured p53 protein translation using a non-radioactive metabolic labeling technique. We think that this approach is better suitable than S35 methionine treatment to measure p53 synthesis in a truly unstimulated state, as genotoxic stress caused by the radioactive labeling itself was shown to result in p53 activation (Yeargin and Haas, 1995). The new data show no difference in p53 synthesis rate in dUTR cells compared with cell containing the WT 3′UTR (new Figure 3C-E).

We used an additional approach to examine p53 production rate. As p53 degradation is dominated by MDM2 (Haupt et al., 1997;; Kubbutat et al., 1997), we examined p53 accumulation by adding increasing concentrations of the MDM2 inhibitor Nutlin-3. We did not detect a significant difference in p53 level in response to Nutlin-3 treatment between parental WT, control clone and dUTR cells (new Figure 3A, 3B), again suggesting that p53 synthesis rate is 3′UTR-independent.

Steady state protein levels are determined by production rate and degradation rate. As our data showed that p53 steady-state as well as production rate were 3′UTR-independent, we did not additionally measure p53 protein stability.

7. The TP53 gene also expresses other p53 isoforms and an antisense transcript called Wrap53. Does deletion of the 3'UTR have any effect on the p53 isoforms and the antisense lncRNA? Addressing this could be important because loss of the p53 3UTR can affect the expression of these genes and profoundly alter the response of cells to DNA damaging agents.

The *WRAP53* gene was shown to regulate p53 expression using an antisense mechanism which is created through an overlap with the 5′ region of the *TP53* gene (Mahmoudi et al., 2009). However, this overlap does not affect the gene region encoding the *TP53* 3′UTR, and both elements are almost 20 kb apart (see genome browser window in Author response image 2). For this reason, we did not consider an effect on p53 expression very likely.

However, we considered the possibility that the *TP53* 3′UTR might be involved in regulating alternative splicing or alternative translation start site usage. Both mechanisms are known to give rise to at least twelve different p53 protein variants (see new Figure 2-Figure supplement 1A). For most analyses in the manuscript, we used a p53 antibody clone (DO-7) that detects all p53 protein variants with a complete N-terminus. In addition, in the revised manuscript, we have added western blot analyses using the p53-specific antibody clone PAb240 that recognizes the p53 DNA binding domain that is present in all isoforms (see new Figure 2-Figure supplement 1A). We did not find evidence for differences in p53 alternative isoform expression as a result of the 3′UTR deletion (new Figure 2-Figure supplement 1B).

8. The authors claim that others have shown regulation of p53 3UTR using reporter assays only. However, endogenous p53 mRNA translation was shown to be regulated by HuR. PMID: 14517280, 12821781.

We did not claim that previous analyses on p53 used exclusively reporter assays. Instead, we listed in Table 1 all experimental approaches that were used by other investigators. However, to avoid confusion, we clarified the text in lines 179 and 245 and say “…often use reporter genes…” or “…has mostly been conducted….”.

When we analyzed existing research on the topic, we evaluated whether an experiment provides direct evidence for binding and regulation by a specific factor, or indirect evidence that can further support that claim. Evidence for direct regulation in *cis* includes assays that abrogate the effect of a factor upon mutation or deletion of the binding site in the 3′UTR. This is usually achieved with reporter constructs, where such manipulations are easily feasible. Alternatively, direct 3′UTR-dependent regulation could be achieved through morpholinos that sterically block access to a specific endogenous binding site, while leaving overall activity of the RBP or miRNA intact as described by Staton and Giraldez (2011). However, none of the studies that investigated p53 regulation used morpholinos.

Most prior papers present indirect evidence in addition to reporter assays. For example, p53 protein levels are measured after overexpression or knockdown of a miRNA or RNA-binding protein. While these experiments likely yield reproducible phenotypes, they cannot distinguish between direct effects on the *TP53* 3′UTR and indirect effects that lead to p53 regulation through other pathways. For example, the RNA-binding proteins HUR and RBM38 have putative binding sites in both *MDM2* and *TP53*, as well as in hundreds of other genes. As a result, measuring p53 protein level as a read-out is not sufficient to unambiguously prove that the regulation occurs in *cis* with the 3′UTR of *TP53* as the main target.

The use of our 3′UTR knock-out strategy enables investigators to tease apart these complex gene regulatory networks. Our study demonstrates for a widely studied and important gene that 3′UTRs do not necessarily regulate mRNA or protein abundance as is often assumed. We have added a more extensive discussion to better explain the potential limitations of prior studies on this subject (lines 283-311).

9. In figure 1a, within the 3'UTR there is a particular segment in which binding sites for multiple RNA-binding proteins overlap. Is there any effect of deleting only this segment?

Given the time that it takes to generate and validate genetically modified cell clones, we were not able to create and analyze new mutant cell lines during the manuscript revision. Instead, we created a reporter construct with this deletion in the *TP53* 3′UTR, called TP53-3′UTR (U-del) that we overexpressed in *TP53-/-* HCT116 cells (new Figure 4C). We created reporter constructs both with and without the *TP53* coding region and measured mRNA and protein levels. We largely confirm the data produced by other groups using very similar constructs (Fu et al., 1999). Specifically, we find that the deletion of the U-rich element leads to de-repression of the reporter mRNA and protein relative to the non-mutated full-length 3′UTR. In fact, this construct yields similar mRNA and protein levels as the much shorter dUTR construct.

Importantly though, when we coupled this *TP53* 3′UTR (U-del) sequence to the *TP53* coding region, these differences disappeared. All constructs that contained the *TP53* coding region (regardless of 3′UTR) were expressed at much lower levels. The combination of the coding region with any of the tested 3′UTR sequences did not result in further repression of the reporter. Our results demonstrate that the *TP53* coding region has a dominant effect on *TP53* mRNA and protein expression. This is a novel insight and we added this result to the abstract to increase the significance of the manuscript.

Reviewer #2:In this interesting manuscript "Endogenous p53 expression in human and mouse is not regulated by its 3′UTR" Mitschka and Mayr examine the gene-regulatory potential of the evolutionarily conserved 3'untranslated region (UTR) of the human and mouse p53 gene (TP53). Using CRISPR-Cas9-mediated gene editing they carefully deleted the entire 3'UTR in human cell lines and mice, leaving processing elements intact. They show that deleting the potential regulatory elements in the TP53 3'UTR does not influence steady state p53 protein expression, as well as p53 protein activation after genotoxic stress.The authors strongly imply that the TP53 3'UTR does not play an important role in TP53 gene function, nevertheless, the conservation of recognizable regulatory elements, such as miRNA binding sites is difficult to square with a lack of function. It would seem necessary to adress this seeming contradiction constructively, considering that it appears to undercut the importance of untranslated regions – if the conserved UTR of a key gene, such as TP53 does not convey some function, are UTRs even important?The presented experiments are unobjectionable and performed to a high standard. That being said, the author's assays, particularly the experiments relating to p53 activation under genotoxic stress mainly cover two aspects: a) translational regulation of the TP53 mRNA and b) posttranslational activation of p53 protein. Considering that the vast majority of the literature points to p53 being activated by genotoxic stress by increasing p53 protein halflife, mainly due to prevention of p53 ubiquitination by MDM2, it is not particularly surprising that once steady-state levels of p53 protein are set, the 3'UTR itself should not play any important role. It is furthermore not surprising that expression of their reporter with the GFP-p53CDS fusion is efficiently suppressed, given that its levels will be controlled by MDM2 in a posttranslational manner. What would happen if steady-state levels of p53 are manipulated, e.g. by simultaneous depletion of MDM2?With p53 being a key protein for cell survival, it is conceivable that its levels are tightly controlled, possibly by multiple buffer systems, not just the UTR alone. E.g. the authors never test whether the half-life of the mRNA itself is impacted – in Figure 1c, they show a trend towards downregulation of steady-state mRNA levels; this in turn could indicate that either transcriptional activity or mRNA half-life are affected. Putting numbers on these parameters might shed light on the presence of additional buffer systems in the cell/mouse ensuring proper p53 protein levels in a sort of fail-safe mechanism.Finally, the authors do not explore the possibility that the effects of their knockouts can be buffered by compensatory mechanisms (e.g. described by the Stainier group that shows that for some genes acute loss by shRNA/siRNAs shows different and more severe phenotypes than gene editing). Of course, this is difficult to test; possibly rescue of full TP53 KO with constructs containing or lacking the TP53 UTR could shed some light on the mRNA dynamics.In summary, by heavily relying on their genotoxic assays that are likely to depend on steady-state p53 protein levels, the authors may have been looking in the wrong place to observe an effect of the TP53 UTR on mRNA dynamics. While their main observation that TP53 UTR deletion by itself does not have a clear phenotype, and miRNA based therapies explored in the context of p53 are likely to fail, they miss the opportunity to address the question whether the TP53 UTR impacts mRNA dynamics and whether the presence of additional buffering mechanisms obscures a direct phenotype.Three major points to be considered:– mRNA halflife and transcription: Putting numbers on these parameters might shed light on the presence of additional buffer systems in the cell/mouse ensuring proper p53 protein levels in a sort of fail-safe mechanism.

We thank the reviewer for spending time to review our manuscript and for the insightful comments. Based on the suggestion of the reviewer, we measured *TP53* mRNA half-life using metabolic labeling of transcripts (see new Figure 1E, 1F) in parental WT, control clones and several dUTR clones. Although we detected a difference in mRNA half-life between the parental cell line and all the clones, we did not observe a 3′UTR-dependent difference in mRNA half-life between the WT and dUTR clones.

– Acute vs chronic loss of p53 UTR: possibly rescue of full TP53 KO with constructs containing or lacking the TP53 UTR could shed some light on the mRNA dynamics.

The data presented in new Figure 4C show experiments in which we overexpressed different p53 constructs in *TP53-/-* HCT116 cells. As such, this experiment represents an acute situation of re-introducing p53 expression from a construct with different 3′UTRs. We found that in the context of the p53 coding region none of the tested 3′UTRs had any influence on *TP53* mRNA or protein output.

The results shown in Figure 4C are crucial to explain the results obtained from the genetically modified cell lines. The data show that the discrepancy between prior studies and our data is not based on the difference between acute vs chronic situations. Instead, these data provide evidence for a dominant repressive effect of the coding region over 3′UTR-mediated regulation with respect to p53 abundance.

– Considering that once p53 steady-state levels are set by the activity of MDM2, the effect of genotoxic stress may be predictable: What would happen if steady-state levels of p53 are manipulated, e.g. by simultaneous depletion of MDM2?

We agree that p53 expression regulation is probably dominated by MDM2 activity. Based on the suggestion of the reviewer, we have now included an experiment in which we added increasing concentrations of the MDM2 inhibitor Nutlin-3 (new Figure 3A, 3B). We did not observe a difference in p53 expression in response to Nutlin-3 treatment between control and dUTR cells. This suggests that MDM2 activity does not mask 3′UTR-mediated p53 expression regulation.

Given that this experiment does not reveal differential accumulation of p53, we did not perform a combined treatment regimen (Nutlin-3 plus genotoxic stress). However, we provide additional data on p53 protein synthesis using metabolic labeling. These experiments were performed in steady state conditions as well as during genotoxic stress. The data shown in the new Figures 3C-E show that p53 synthesis (in unstressed or stress conditions) does not depend on the 3′UTR. The data added to the revised manuscript rule out additional buffering mechanisms that obscure a 3′UTR-dependent effect on p53 abundance regulation.

Reviewer #3:[…] In general, this is a nicely articulated manuscript with interesting data. The data argue against the role of 3'UTR in the differential expression of p53. However the significance of the findings are not entirely clear to this reviewer, and it is felt that the authors may need to perform experiments to test the role of other elements (5'UTR as well as coding region) in regulating p53 stability as well as translation.1. The steady state levels of a protein is dictated by regulated translation versus protein degradation, and each of these processes are controlled by distinct sequence elements located within the transcript. Does the 3'UTR of p53 influence differential translation? The authors really should check the polysome association of WT as well dUTR p53 mRNA.

We thank the reviewer for spending time to review our manuscript and for the insightful comments. Based on the suggestion of the reviewer, we examined the role of the *TP53* 3′UTR in the regulation of translation. Instead of using polysome association to measure translation, we used metabolic labeling with the non-radioactive methionine analog AHA to measure protein synthesis. The measurement of newly synthesized protein is a more direct read-out for mRNA translation than polysome association. This experiment showed that p53 protein synthesis is independent of the 3′UTR as we did not detect significant differences in HCT116 derived WT parental, control clones, and dUTR cells in both steady state conditions and during genotoxic stress (new Figures 3C-E).

Moreover, we measured protein synthesis as a function of p53 accumulation when degradation was pharmacologically inhibited using the MDM2 inhibitor Nutlin-3. Also with this method we did not observe 3′UTR-dependent differences in p53 protein synthesis rate (new Figure 3A, 3B).

2. Both the WT and the 3'UTR present in the dUTR constructs contain a small region, which based on earlier studies, can support the binding of a few RBPs and miRNAs. It cannot be completely ruled out that those interactions might influence the post transcriptional processing of the p53 mRNA. It might be worth doing an experiment in which the entire 3'UTR of the endogenous p53 could be replaced with the 3'UTR of a house-keeping gene to see the effect on p53 mRNA stability, polysome association and translation.

The genomic 3′UTR deletion was designed to preserve endogenous 3′ end processing, which is essential to produce an mRNA through cleavage and polyadenylation. We would like to refer the reviewer to our response to Rev1Q1 for a detailed explanation on the criteria we used to determine the relevant regions.

It is correct that a small number of potential regulatory sites lie outside the deleted 3′UTR region. In order to address the question if the remaining 3′UTR sequence is actively involved in p53 expression regulation, we expanded our set of test 3′UTRs to include three 3′UTRs of housekeeping genes (*GAPDH*, *HPRT*, *PGK1*) in the reporter assay where we measured reporter mRNA and protein expression (new Figure 4C). We also added a reporter construct using the BGH poly(A) terminator sequence that generates a 3′UTR with a length of 225 nucleotides. This construct serves as a baseline for reporter gene expression. When comparing reporter gene expression between the three housekeeping gene 3′UTRs and the dUTR construct, we consistently observed lower mRNA and protein expression from the housekeeping gene 3′UTRs (new Figure 4C). We conclude that the remaining sequence in the dUTR construct is not repressive.

It is important to highlight again that any 3′UTR-dependent differences (between the three housekeeping gene 3′UTRs and the dUTR fragment) disappeared in the context of the p53 coding region (new Figure 4C). This result demonstrates that the coding region has a dominant repressive effect on *TP53* mRNA and protein expression.

Furthermore, we added the 5′UTR to the reporter (Figure 4-Figure supplement 1B) to test the contribution of the 5′UTR to p53 expression regulation in steady state conditions. In the presence of the *TP53* coding region, the 5′UTR did not have any additional effect on reporter mRNA or protein expression.

3. The reporter assays indicated that both the 3'UTR as well as the CDS region contribute to the steady state levels of p53 mRNA. It is not clear why the dUTR alone containing reporter showed increased steady state levels of reporter mRNA compared to WT, if the 3'UTR of p53 does not have any function in regulating the stability of p53 mRNA. This needs to be better-explained.

We understand the non-intuitive nature of this result. Although we are not the first to observe clear evidence for sequence context-dependent expression regulation (Cottrell et al., 2017;; Theil et al., 2018), the phenomenon remains poorly conceptualized. In the ‘Discussion’ of the revised manuscript (lines 257-283), we are suggesting three potential mechanisms that could explain these data:

1. Adding more sequence context could alter mRNA structure and impede accessibility to specific sequence motifs.

2. Effect saturation could be achieved if the pathways that are activated by different *trans*-acting factors converge on certain downstream effector complexes.

3. Recruitment of different *trans*-acting factors in a combinatorial manner could enable new epistatic effects.

**References:**

Cottrell, K. A., Szczesny, P., and Djuranovic, S. (2017). Translation efficiency is a determinant of the magnitude of miRNA-mediated repression. Sci Rep, 7(1), 14884.

Fu, L., Ma, W., and Benchimol, S. (1999). A translation repressor element resides in the 3' untranslated region of human p53 mRNA. Oncogene, 18(47), 6419-6424.

Haupt, Y., Maya, R., Kazaz, A., and Oren, M. (1997). Mdm2 promotes the rapid degradation of p53. Nature, 387(6630), 296-299.

Kubbutat, M. H., Jones, S. N., and Vousden, K. H. (1997). Regulation of p53 stability by Mdm2. Nature, 387(6630), 299-303.

Mahmoudi, S., Henriksson, S., Corcoran, M., Mendez-Vidal, C., Wiman, K. G., and Farnebo, M. (2009). Wrap53, a natural p53 antisense transcript required for p53 induction upon DNA damage. Mol Cell, 33(4), 462-471.

Martin, G., Gruber, Andreas R., Keller, W., and Zavolan, M. (2012). Genome-wide Analysis of Pre-mRNA 3ʹ′ End Processing Reveals a Decisive Role of Human Cleavage Factor I in the

Regulation of 3ʹ′ UTR Length. Cell Reports, 1(6), 753-763.

Staton, A. A., and Giraldez, A. J. (2011). Use of target protector morpholinos to analyze the physiological roles of specific miRNA-mRNA pairs in vivo. Nat Protoc, 6(12), 2035-2049.

Theil, K., Herzog, M., and Rajewsky, N. (2018). Post-transcriptional Regulation by 3' UTRs Can Be Masked by Regulatory Elements in 5' UTRs. Cell Rep, 22(12), 3217-3226.

Yeargin, J., and Haas, M. (1995). Elevated levels of wild-type p53 induced by radiolabeling of cells leads to apoptosis or sustained growth arrest. Curr Biol, 5(4), 423-431.